# Do Remittances Promote Economic Growth and Reduce Poverty? Evidence from Latin American Countries

## E. M. Ekanayake [1,*] and Carlos Moslares [2]

1   College of Business and Entrepreneurship, Bethune-Cookman University, Daytona Beach, FL 32114, USA
2   IQS School of Management, Universitat Ramon Llull, Via Augusta, 390, 08017 Barcelona, Spain; carles.moslares@iqs.edu
*   Correspondence: ekanayakee@cookman.edu; Tel.: +1-386-481-2819

**Abstract:** In this study, we explore the hypotheses that (a) workers' remittances enhance economic growth in Latin American countries, and (b) workers' remittances help reduce poverty in Latin American countries. In recent decades, workers' remittances have become an important source of income for many developing countries and, as a global aggregate, workers' remittances are the largest source of foreign financing after foreign direct investment. This paper analyzes the effects of workers' remittances on economic growth and poverty in 21 Latin American countries. The study uses annual data covering all Latin American countries for the period 1980–2018. We employ panel least squares and panel fully-modified least squares (FMOLS) methods. In addition, we estimate the short-run and long-run effects of workers' remittances on economic growth and poverty on individual countries with the Autoregressive Distributed Lag (ARDL-ECM) approach to co-integration analysis. The results reveal that workers' remittances have a positive effect on long-run economic growth in the majority of the countries studied, but have mixed effects in the short-run. They also suggest that workers' remittances tend to lower poverty rates in Latin America.

**Keywords:** remittances; economic growth; poverty; Latin America; ARDL; FMOLS

---

## 1. Introduction

In recent decades, remittances have become an important source of foreign capital for many developing countries. However, there is no general consensus in policy debates on the impact of remittances on economic growth and poverty, despite the increasing reliance of developing countries on private capital flows as a main source of funding. Remittances are defined as the sum of workers' remittances, compensation of employees, and migrant transfers. According to the World Bank (2019c), worldwide flows of remittances to the developing world reached US$583 billion in 2017 and US$624.5 billion in 2018. Remittances sent home by migrants from developing countries have maintained a steady and marked upward trend between 1980 and 2018, reaching US$424.2 billion in 2018 compared to US$12.0 billion in 1980 (see Table 1). Recorded remittances are more than twice as large as official aid and nearly two-third of foreign direct investment flows to developing countries. Remittances are the largest source of external financing in many poor countries. Workers' remittances have in fact become the second most important type of private external finance to developing countries after FDI.

**Table 1.** Global Remittances Flows, 1980–2018.

| | Remittances (US$ Billions) | | | | | Share of Remittances (%) | | | | |
|---|---|---|---|---|---|---|---|---|---|---|
| **Region** | **1980** | **1990** | **2000** | **2010** | **2018** | **1980** | **1990** | **2000** | **2010** | **2018** |
| All Developing Countries | 12.0 | 17.5 | 52.7 | 268.4 | 424.2 | 32.3 | 25.5 | 43.3 | 64.1 | 67.9 |
| Low Income | 1.3 | 2.4 | 2.8 | 14.4 | 27.1 | 3.6 | 3.5 | 2.3 | 3.4 | 4.3 |
| Lower Middle Income | 10.7 | 15.7 | 39.2 | 180.5 | 288.0 | 28.9 | 23.0 | 32.2 | 43.1 | 46.1 |
| Middle Income | 16.4 | 27.0 | 71.4 | 287.8 | 452.2 | 44.3 | 39.5 | 58.7 | 68.8 | 72.4 |
| Upper Middle Income | 5.7 | 11.3 | 32.2 | 107.2 | 164.3 | 15.3 | 16.5 | 26.5 | 25.6 | 26.3 |
| East Asia and Pacific | 2.6 | 8.7 | 18.7 | 68.8 | 115.3 | 7.1 | 12.6 | 15.4 | 16.4 | 18.5 |
| Europe and Central Asia | 2.1 | 3.2 | 8.7 | 37.9 | 55.0 | 5.6 | 4.7 | 7.2 | 9.1 | 8.8 |
| Latin America and Carib. | 1.9 | 5.7 | 19.8 | 57.0 | 89.9 | 5.2 | 8.4 | 16.3 | 13.6 | 14.4 |
| Middle-East and N. Africa | 6.5 | 10.5 | 11.6 | 38.2 | 59.7 | 17.6 | 15.3 | 9.5 | 9.1 | 9.6 |
| South Asia | 5.3 | 5.6 | 17.2 | 82.0 | 131.1 | 14.3 | 8.1 | 14.1 | 19.6 | 21.0 |
| Sub-Saharan Africa | 1.4 | 2.4 | 4.8 | 31.6 | 46.7 | 3.8 | 3.5 | 3.9 | 7.5 | 7.5 |
| *High Income OECD* | *22.2* | *45.0* | *58.2* | *133.2* | *172.7* | *59.9* | *65.8* | *47.8* | *31.8* | *27.7* |
| High Income non-OECD | 2.9 | 6.0 | 10.8 | 16.9 | 27.5 | 7.7 | 8.7 | 8.9 | 4.0 | 4.4 |
| World | 37.0 | 68.4 | 121.6 | 418.5 | 624.5 | 100.0 | 100.0 | 100.0 | 100.0 | 100.0 |

Source: World Bank, *World Development Indicators 2019 Database*.

Remittance flows to Latin America have also dramatically increased in the past several decades to become a major force in the worldwide allocation of funds and technology. Between 1980 and 2018, remittance flows to Latin America and the Caribbean increased 47-fold from US$1.9 billion in 1980 to US$89.9 billion in 2018, though its share in world remittance flows has only increased from 5.2 percent to 14.4 percent during this period (see Table 1). It is also important to note that, on average, Mexico has received nearly half of the total remittance flows to Latin America. According to the World Bank (2019a), Mexico continued to receive the most remittances in the region, posting about $36 billion in 2018, up 11 percent over the previous year, supported by the strong U.S. economy. Colombia and Ecuador, both of which have migrants in Spain, posted 16 percent and 8 percent growth, respectively, in 2018. Three other countries in the Latin America and the Caribbean region posted double-digit growth in 2018: Guatemala (13 percent) as well as the Dominican Republic (10 percent) and Honduras (10 percent), reflecting robust outbound remittances from the United States. According to Table 2, Mexico's share in Latin American remittances dropped from 60.1 percent in 1980 to 42.5 percent in 2018, with a period average of 49.4 percent during 1980–2018. Nonetheless, remittances to Mexico grew at an annual average rate of 10.3 percent during this period, somewhat slower than the annual average growth rate for the entire region (11.1 percent). Of the countries that are presented in Table 2, the majority experienced significant growth in remittances flows during the 1980–2018 period. Of the 21 Latin American countries included in Table 2, nine of them (Argentina, Belize, Bolivia, Chile, Costa Rica, Guyana, Suriname, Uruguay, and Venezuela) individually received less than 1 percent of total remittances to Latin America during the 1980–2018 period. According to the United Nations Development Program (UNDP), remittances make several contributions to economic development, including: (a) they could become an important tool for economic development if channeled into productive investment; (b) from a macroeconomic perspective, they can generate output growth either by increasing consumption or by increasing investment; and (c) remittances increase the ability of households to spend on health, housing, and nutrition that can enhance their productivity and spur economic growth over the longer term.

**Table 2.** Remittances Flows to Latin America, 1980–2018.

| Country | Remittances Flows to Latin America (US$ Millions) | | | | | | | | 1980–2018 Annual Avg. Growth (%) |
|---|---|---|---|---|---|---|---|---|---|
| | 1980 | 1985 | 1990 | 1995 | 2000 | 2005 | 2010 | 2018 | |
| Argentina | 56.0 | 27.0 | 22.8 | 63.6 | 86.3 | 432.1 | 644.3 | 507.5 | 12.0 |
| Belize | 15.0 | 21.0 | 18.5 | 13.9 | 24.9 | 44.7 | 78.1 | 92.5 | 6.0 |
| Bolivia | 1.4 | 6.0 | 4.6 | 5.4 | 126.9 | 337.0 | 960.2 | 1391.6 | 39.5 |
| Brazil | 111.0 | 40.0 | 573.0 | 2952.0 | 1349.6 | 2805.4 | 3082.8 | 2933.5 | 22.4 |
| Chile | 1.0 | 1.0 | 0.4 | 1.2 | 13.3 | 13.0 | 62.3 | 66.1 | 15.9 |
| Colombia | 106.0 | 110.0 | 495.0 | 815.1 | 1,610.1 | 3,345.6 | 4030.8 | 6367.5 | 18.4 |
| Costa Rica | 4.1 | 7.2 | 12.0 | 123.3 | 136.0 | 420.3 | 530.7 | 533.5 | 26.4 |
| Dom. Rep. | 183.1 | 242.0 | 314.8 | 839.2 | 1838.8 | 2719.2 | 3887.0 | 6814.2 | 11.1 |
| Ecuador | 1.0 | 2.0 | 51.0 | 386.1 | 1322.3 | 2460.0 | 2599.0 | 3039.1 | 16.0 |
| El Salvador | 49.0 | 157.2 | 366.3 | 1063.9 | 1764.2 | 3028.6 | 3471.8 | 5388.1 | 14.1 |
| Guatemala | 26.2 | 1.0 | 118.7 | 357.5 | 596.2 | 3066.6 | 4231.8 | 9490.6 | 9.1 |
| Guyana | 1.2 | 1.0 | 1.0 | 1.7 | 27.3 | 201.3 | 367.8 | 285.5 | 35.3 |
| Honduras | 1.6 | 2.1 | 62.9 | 124.0 | 474.5 | 1805.2 | 2617.9 | 4776.5 | 56.0 |
| Mexico | 1039.0 | 1616.0 | 3098.0 | 4368.1 | 7524.7 | 22741.8 | 22080.3 | 35561.6 | 10.3 |
| Nicaragua | 5.0 | 6.5 | 8.5 | 75.0 | 320.0 | 615.7 | 824.8 | 1,504.8 | 18.7 |
| Panama | 65.2 | 99.0 | 109.7 | 112.0 | 16.4 | 129.6 | 410.0 | 537.8 | 14.4 |
| Paraguay | 52.1 | 9.6 | 33.8 | 134.9 | 151.7 | 161.3 | 409.9 | 682.9 | 11.4 |
| Peru | - | - | 87.0 | 599.0 | 717.7 | 1440.1 | 2533.9 | 3224.8 | 15.3 |
| Suriname | 6.0 | 4.5 | 0.5 | 0.3 | 1.3 | 3.9 | 4.3 | 0.5 | 18.7 |
| Uruguay | 4.0 | 6.5 | 10.4 | 16.8 | 27.1 | 76.7 | 124.9 | 104.3 | 9.7 |
| Venezuela | 1.0 | 1.0 | 1.0 | 2.0 | 17.0 | 148.0 | 143.0 | 297.1 | 42.1 |
| Latin America | 1728.9 | 2360.6 | 5389.9 | 12,055.0 | 18,146.2 | 45,996.2 | 53,095.7 | 83,599.9 | 11.1 |

| Country | Share of Remittances Flows to Latin America (%) | | | | | | | | 1980–2018 Annual Avg. Share (%) |
|---|---|---|---|---|---|---|---|---|---|
| | 1980 | 1985 | 1990 | 1995 | 2000 | 2005 | 2010 | 2018 | |
| Argentina | 3.2 | 1.1 | 0.4 | 0.5 | 0.5 | 0.9 | 1.2 | 0.6 | 0.9 |
| Belize | 0.9 | 0.9 | 0.3 | 0.1 | 0.1 | 0.1 | 0.1 | 0.1 | 0.3 |
| Bolivia | 0.1 | 0.3 | 0.1 | 0.0 | 0.7 | 0.7 | 1.8 | 1.7 | 0.8 |
| Brazil | 6.4 | 1.7 | 10.6 | 24.5 | 7.4 | 6.1 | 5.8 | 3.5 | 7.4 |
| Chile | 0.1 | 0.0 | 0.0 | 0.0 | 0.1 | 0.0 | 0.1 | 0.1 | 0.1 |
| Colombia | 6.1 | 4.7 | 9.2 | 6.8 | 8.9 | 7.3 | 7.6 | 7.6 | 8.1 |
| Costa Rica | 0.2 | 0.3 | 0.2 | 1.0 | 0.7 | 0.9 | 1.0 | 0.6 | 0.7 |
| Dom. Rep. | 10.6 | 10.3 | 5.8 | 7.0 | 10.1 | 5.9 | 7.3 | 8.2 | 7.9 |
| Ecuador | 0.1 | 0.1 | 0.9 | 3.2 | 7.3 | 5.3 | 4.9 | 3.6 | 3.2 |
| El Salvador | 2.8 | 6.7 | 6.8 | 8.8 | 9.7 | 6.6 | 6.5 | 6.4 | 7.0 |
| Guatemala | 1.5 | 0.0 | 2.2 | 3.0 | 3.3 | 6.7 | 8.0 | 11.4 | 4.7 |
| Guyana | 0.1 | 0.0 | 0.0 | 0.0 | 0.2 | 0.4 | 0.7 | 0.3 | 0.2 |
| Honduras | 0.1 | 0.1 | 1.2 | 1.0 | 2.6 | 3.9 | 4.9 | 5.7 | 2.6 |
| Mexico | 60.1 | 68.5 | 57.5 | 36.2 | 41.5 | 49.4 | 41.6 | 42.5 | 49.4 |
| Nicaragua | 0.3 | 0.3 | 0.2 | 0.6 | 1.8 | 1.3 | 1.6 | 1.8 | 1.0 |
| Panama | 3.8 | 4.2 | 2.0 | 0.9 | 0.1 | 0.3 | 0.8 | 0.6 | 1.4 |
| Paraguay | 3.0 | 0.4 | 0.6 | 1.1 | 0.8 | 0.4 | 0.8 | 0.8 | 1.0 |
| Peru | - | - | 1.6 | 5.0 | 4.0 | 3.1 | 4.8 | 3.9 | 2.9 |
| Suriname | 0.3 | 0.2 | 0.0 | 0.0 | 0.0 | 0.0 | 0.0 | 0.0 | 0.1 |
| Uruguay | 0.2 | 0.3 | 0.2 | 0.1 | 0.1 | 0.2 | 0.2 | 0.1 | 0.2 |
| Venezuela | 0.1 | 0.0 | 0.0 | 0.0 | 0.1 | 0.3 | 0.3 | 0.4 | 0.2 |
| Latin America | 100.0 | 100.0 | 100.0 | 100.0 | 100.0 | 100.0 | 100.0 | 100.0 | 100.0 |

Source: World Bank, *World Development Indicators 2019 Database*.

Figure 1 illustrates the trends in worker remittances, poverty rate, and economic growth rate in Latin America during the period from 1980 to 2018. The poverty rate is expressed as a simple average of poverty rates, which is measured using the global measure of extreme poverty ($1.90 international dollars (2011) per person per day). Similarly, economic growth rate of the region is also expressed as a simple average of economic growth rates of each country. As the figure illustrates, the poverty rate has declined consistently since 2000. Some of the factors that contributed to this decline include higher spending on education, improved healthcare access, and improve infrastructure (Vacaflores 2018). In addition, the redistributive efforts of the left-leaning administrations in the region may have also contributed to the declining poverty and improved standards of living in the region. As Vacaflores (2018) points out, worker remittances could improve the standard of living of the poorest segments of the population, since it allows them to increase their consumption and to start new businesses. While the average economic growth rate in the region has remained relatively low during the study period, it shows a declining trend since 2010. This decline happened when the region experienced a significant increase in remittances flows.

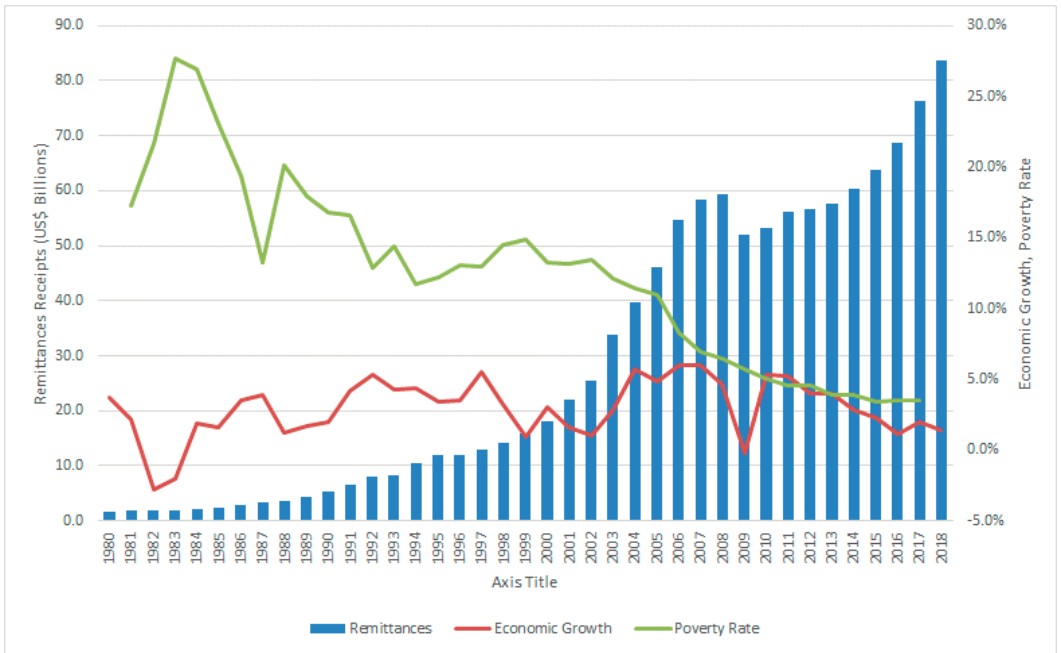

**Figure 1.** Trends in Remittances, Economic Growth, and Poverty in Latin America, 1980–2018. Note: Authors' own calculation using data from the World Bank, *World Development Indicators 2019 Database* and from the *Socio-Economic Database for Latin America and the Caribbean (SEDLAC)*. The poverty rate is expressed as a simple average of poverty rates which is measured using the global measure of extreme poverty ($1.9 international dollars per person per day).

Data on the gross domestic product (GDP) and the poverty rate for 21 Latin American countries are presented in Appendix B Tables A1 and A2. The majority of the countries in Latin America are relatively small economies, with 13 of the 21 countries reporting a GDP less than US$100 billion in 2018 (see Appendix B Table A1). Brazil and Mexico are the two largest economies in the region. Regardless of their smaller size, majority of the countries in the region has maintained a moderate level of economic growth. Data on poverty rates are missing for several countries in our selected sample of countries, as Appendix B Table A2 illustrates. Based on the available statistics, we can conclude that the poverty rates in Latin America have declined during the period from 2000 to 2017. As Table A2 in the Appendix B shows, poverty rates in Argentina, Bolivia, Brazil, Colombia, Costa Rica, Dominican Republic, El Salvador, Honduras, Mexico, Panama, Paraguay, and Uruguay have declined significantly during this time period.

The paper is structured as follows: The next section presents a survey of the literature, while Section 3 specifies the econometric model and data sources. The empirical results are presented and discussed in Section 4 and finally, Section 5 summarizes the main results and concludes with some policy implications.

## 2. Literature Review

The relationships between remittances and economic growth and between remittances and poverty have drawn great attention in recent years; however, the literature is ambivalent on the nature of these relationships. Some studies have found evidence to suggest that remittances promote economic growth and lower poverty, while others found evidence to suggest that remittances have a negative effect on economic growth. Though there are a large number of studies on the remittances-growth nexus, for this review we have selected the few recent studies that cover Latin America and the Caribbean (LAC) region. As pointed out by Glytsos (2005), considering the dependence of remittance flows on complex factors related to the nature and purpose of migration, the changing migrant flows entail complex and multidimensional effects of remittances, which make their role difficult to detect and evaluate.

Giuliano and Ruiz-Arranz (2009) found that remittances have a positive and significant impact on economic growth, while Mundaca (2009) also found positive effects for a sample of countries covering the LAC region. Giuliano and Ruiz-Arranz (2009) used a sample of 100 developing countries with annual data for the period 1975–2002 and reported results for both ordinary least squares (OLS) and System Generalized Method of Moments (SGMM) regressions. The study concluded that remittances have a positive and significant effect on economic growth and financial development facilitates such growth (especially in countries with less developed financial systems), but have a negative impact in countries with a more developed financial sector. The study by Mundaca (2009) assessed the impact of remittances on growth using a sample of 25 LAC countries and found a strong positive correlation between remittances and economic growth, and that the impact was stronger when the financial sector was included in the model. Nsiah and Fayissa (2013) showed that there is a long-run relationship between per capita remittances and per capita income in a sample of 21 LAC countries and that the relationship between the two variables is positive. Ramirez and Sharma (2008) also provided evidence for the long-run relationship between per capita GDP growth and the remittance to GDP ratio in a sample of 23 LAC countries for the 1990–2005 period. They showed that the impact is greater in countries with less access to private credits. Pradhan et al. (2008) examined the effects of workers' remittances on economic growth using panel data for 39 developing countries covering the period from 1980 to 2004 concluded that remittances have a positive effect on economic growth. World Bank (2006a; 2006b) and Fajnzylber and López (2007) also established that the growth impact of remittances in the LAC region is positive, though the magnitude is relatively small. Fajnzylber and López (2007, 2008) concluded that remittances seem to accelerate growth rates and reduce poverty levels in Latin America, and one potential channel that has been highlighted in their studies is the impact of these flows on financial sector development, savings, and investment. Other studies that have found a positive relationship between remittances and economic growth include Catrinescu et al. (2009); Faini (2007); and Ziesemer (2006). Chami et al. (2005) found that remittances have a negative impact on economic growth, arguing that they are compensatory transfers and hence are countercyclical. The study uses a panel of aggregate data covering 113 countries and 29 years and contends that remittances are intended for consumption and do not act like a source of capital for economic development. The fact that remittances are initially spent on consumption, housing, and land, and are not used for direct productive investment is often taken as a loss of resources for promoting long-term growth and development. Some other studies that found a negative or no relation between remittances and economic growth include Barajas et al. (2009); Gupta (2005); and the International Monetary Fund (IMF). Lim and Simmons (2015) investigated the economic importance of remittances flows to 13 countries in the Caribbean Community and Common Market (CARICOM), using the data covering the 1975–2010 period. The study was unable to find any evidence for a long-run association between

remittances and real GDP per capita in the region. The study concluded that the remittances flows to the Caribbean are mostly used to finance consumption needs rather than investing in growth-enhancing projects. For a recent comprehensive survey of the theoretical and empirical literature on remittances on economic growth in developing countries, see Ziesemer (2012).

A fairly large number of papers have studied the questions of the impact of remittances on poverty and income distribution, while the econometric work on the effects of remittances is rather extensive (see, for example: Vacaflores 2018; Akobeng 2016; Imaia et al. 2014; Adams and Cuecuecha 2013; Gupta et al. 2009; Adams and Page 2003, 2005; Rivera-Batiz 1986; Lundahl 1985). Empirical evidence on the effects of remittances on poverty is also mixed and there is little consensus in the literature concerning the impact of worker remittances on poverty. Using a dataset for 18 Latin American countries covering the 2000–2013 period, Vacaflores (2018) examined the effectiveness of international remittances in reducing poverty and inequality, and concluded that the increases in remittances have a negative and statistically significant impact on overall poverty and inequality in the region. Akobeng (2016) used data for a group of 41 Sub-Saharan African countries to investigate the impact of international remittances on inequality and poverty and found that remittances reduce poverty, though the impact depends on how poverty is being measured. Gupta et al. (2009) investigated the effect on remittances on poverty and financial development in Sub-Saharan Africa with the inclusion of 24 Sub-Saharan countries, and concluded that remittances reduce poverty and promote financial development. Adams and Page (2003, 2005) have shown that high levels of remittances are associated with lower poverty indicators. Acosta et al. (2008) used data for 59 industrial and developing countries spanning the years 1970–2000 based on the Penn World Tables (PWT) 6.1 database and showed evidence that remittances lower poverty. In order to separate the LAC countries from the rest of the countries, the study introduced a dummy variable. Using a household survey for Latin American countries, Acosta et al. (2008) found that remittances have negative and relatively small inequality and poverty-reducing effects. Acosta et al. (2006) showed that remittances reduce poverty headcounts but do not reduce inequality in this region. Using a dataset of 149 countries, Cattaneo (2005) found that remittances do not have any impact on poverty. Adams and Cuecuecha (2013) used a data set covering 71 developing countries to investigate the impact of international migration and remittances on poverty, and concluded that both international migration and remittances have a strong, statistically significant impact on poverty reduction in developing countries.

The motivation to conduct this study arose because there is no general consensus in policy debates on the impact of remittances on economic growth and poverty, and because the number of studies that have examined these issues in the Latin American region is relatively small. Considering the growing economic importance of remittance flows to Latin American region, this paper attempts to fill this gap in empirical research. This paper employs panel least squares and panel fully-modified least squares (FMOLS) methods to estimate the effects of workers' remittances on economic growth and poverty in 21 Latin American countries using a newly available dataset. In addition, we also employ bounds testing or the Autoregressive Distributed Lag (ARDL) approach to co-integration analysis to empirically assess the effects of remittance flows on the economic growth in individual countries. The paper also assesses the role of the institutions in determining the relative effectiveness of remittance flows to the region. The specific objectives of this study are to explore the hypotheses that (a) workers' remittances will enhance economic growth in Latin American countries, and (b) workers' remittances will help reduce poverty levels in Latin American countries. These hypotheses are tested at two levels. First, we explore the effects of remittances on economic growth and poverty when we consider all countries as a group. In the second step, we explore the effects of remittances on economic growth for each individual country. We were unable to conduct a similar analysis at the individual country level due to the lack of data on poverty. One of the innovations of this paper is to analyze the effects of remittances on economic growth at the individual country level. Such an analysis has not been undertaken by any previous study. Another innovation of the paper is that it has incorporated the institutional environment, which is vital for enhancing growth and the development

impact of remittances in Latin America. Thus, the findings reported in this study represent a significant contribution to the existing literature, particularly because they have been derived using recently developed econometric techniques and a larger dataset.

Unlike the studies reviewed in this section, our approach deals more specifically with 21 Latin American countries, covering a relatively longer period of time 1980–2018 and utilizing panel least squares and fully-modified least squares (FMOLS) and method of bounds testing or the Autoregressive Distributed Lag (ARDL) approach to co-integration analysis.

## 3. Methodology

### 3.1. Specification of Models

The estimated model was derived from a standard growth equation based on a traditional production function. Worker remittances are a prime source of revenue to developing countries and this variable was included in the growth equation and poverty equation in order to capture its role.

In the usual notation the production function can be written as follows:

$$Y_t = f(L_t, K_t, HC_t) \tag{1}$$

where $Y$ is real gross domestic product (*GDP*) in year $t$, $L$ is the number of workers employed in the economy, $K$ is the amount of capital input (measured in physical units or in $ value) in the economy, and $HC$ is the level of human capital in the economy. Differentiating Equation (1) and dividing it by $Y_t$, we obtain the following equation:

$$\frac{\Delta Y_t}{Y_t} = \frac{\Delta Y}{\Delta L}\frac{\Delta L_t}{Y_t} + \frac{\Delta Y}{\Delta K}\frac{\Delta K_t}{Y_t} + \frac{\Delta Y}{\Delta HC}\frac{\Delta HC_t}{Y_t} \tag{2}$$

$$\frac{\Delta Y_t}{Y_t} = \frac{\Delta Y}{\Delta L}\frac{\Delta L_t}{Y_t}\frac{L_t}{L_t} + \frac{\Delta Y}{\Delta K}\frac{\Delta K_t}{Y_t}\frac{K_t}{K_t} + \frac{\Delta Y}{\Delta HC}\frac{\Delta HC_t}{Y_t}\frac{HC_t}{HC_t} \tag{3}$$

$$\frac{\Delta Y_t}{Y_t} = \frac{\Delta Y}{\Delta L}\frac{L_t}{Y_t}\frac{\Delta L_t}{L_t} + \frac{\Delta Y}{\Delta K}\frac{K_t}{Y_t}\frac{\Delta K_t}{K_t} + \frac{\Delta Y}{\Delta HC}\frac{HC_t}{Y_t}\frac{\Delta HC_t}{HC_t} \tag{4}$$

$$\frac{\Delta Y_t}{Y_t} = \theta_L\frac{\Delta L_t}{L_t} + \theta_K\frac{\Delta K_t}{K_t} + \theta_{HC}\frac{\Delta HC_t}{HC_t} \tag{5}$$

$$g_Y = \theta_L g_L + \theta_K g_K + \theta_{HC} g_{HC} \tag{6}$$

where $g_Y$, $g_L$, $g_K$, and $g_{HC}$ are growth rates of $Y$, $L$, $K$, and $HC$, respectively and $\theta_L$, $\theta_K$, and $\theta_{HC}$ are output elasticities of $L$, $K$, and $HC$, respectively.

After adding a constant term to growth Equation (6), we obtained the following expression describing the determinants of the growth rate of real GDP:

$$g = \alpha + \beta\, l + \gamma\, k + \delta\, hc \tag{7}$$

where $g$ is the growth rate of real per capita GDP, $l$ is the growth rate of labor, $k$ is the growth rate of capital, and $hc$ is the growth rate of human capital. To estimate the links between remittances and economic growth, we modified this growth equation by adding the growth rate of remittances as an additional independent variable. Following the precedent set in numerous previous studies, we approximated the rate of growth of the capital stock by the share of investment in GDP. This is necessary due to the formidable problems associated with attempts to measure the capital stock, especially in the context of developing countries. Though Penn World Table 9.0 has data on real capital stock, such data are not available fuor some of the countries included in this study and therefore we decided to use the share of investment in GDP for all countries in our sample. We used the share of the population with a minimum edcational attainment level (Bachelor's or equivalent) out of the broader population that is 25 years old and older as a measure of human capital. In addition, we also

replaced the rate of change in labor input by the growth rate of labor force and expressed the real inflow of workers' remittances as a percentage of GDP. We used the growth rate of real per capita GDP as the measure of economic growth. Following Acosta et al. (2006); Giuliano and Ruiz-Arranz (2009); and Chami et al. (2005), we also introduced the lagged real per capita GDP as one of the explanatory variables. Following Acosta et al. (2006), we also used lagged remittances. These changes yielded the following growth equation:

$$g_{it} = \mu_i + \delta_i t + \beta_1 y_{i(t-1)} + \beta_2 l_{it} + \beta_3 k_{it} + \beta_3 hc_{it} + \beta_5 wr_{i(t-1)} + \omega_{it} \tag{8}$$

where $g$ is the growth rate of real per capita GDP, $y$ is the log of real per capita GDP, $l$ is the log of growth rate of labor force, $k$ is the log of growth rate of capital stock as proxied by the investment to GDP ratio, hc is the log of growth rate of human capital, wr is the log of workers' remittance to GDP ratio, $i$ = 1, 2, 3, .., 21 for each Latin American country in the panel and $t$ = 1, 2, 3, ...., 39 refers to the time period (from 1980 to 2018). The parameters $\mu_i$ and $\delta_i$ allow for country-specific fixed-effects and deterministic trends, respectively, while $\omega_{it}$ denote the estimated residuals which represent deviations from the long-run relationship. The dependent variable represents the growth rate of real per capita GDP. According to economic theory, the expected sign of the parameter $\beta_1$ is negative, the expected sign of the parameter $\beta_2$ is positive, the expected sign of the parameter $\beta_3$ is positive, the expected sign of the parameter $\beta_4$ is positive, the expected sign for parameter $\beta_5$ can expected to be either positive or negative depending on the impact of workers' remittances on the economy.

Following the approach in Gupta et al. (2009) and Adams and Page (2003), we specify the following empirical model specify the model to test the relationship between remittances and poverty.

$$pov_{it} = \alpha_i + \theta_i t + \beta_1 pov_{i(t-1)} + \beta_2 y_{it} + \beta_3 wr_{i(t-1)} + \varepsilon_{it} \tag{9}$$

where $pov_{it}$ is the log of poverty rate in country i in period t, $i$ = 1, 2, 3, .., 21 for each Latin American country in the panel and $t$ = 1, 2, 3, ...., 39 refers to the time period, and the other two variables were defined earlier in Equation (8). The parameters $\alpha_i$ and $\theta_i$ allow for country-specific fixed-effects and deterministic trends, respectively, while $\varepsilon_{it}$ denote the estimated residuals which represent deviations from the long-run relationship. The expected sign of the parameter $\beta_1$ is positive, the expected sign of the parameter $\beta_2$ is negative, and the expected sign for parameter $\beta_3$ can expected to be either positive or negative depending on the impact of workers' remittances on poverty.

As Catrinescu et al. (2009) pointed out, the effects of remittances on economic growth could work through several different channels with institutions being one of the most important channels. Given this importance, we have also introduced several institutional variables into Equations (8) and (9). Introduction of institutional variables limited the number of years to 23 years, from 1996 to 2018, since the information on these variables is available only from 1996. To measure institutions, we used six Worldwide Governance Indicators from the World Bank (2018). These indicators measure six dimensions of governance: Voice and Accountability, Political Stability and Absence of Violence, Government Effectiveness, Regulatory Quality, Rule of Law, and Control of Corruption. The models specified in Equations (8) and (9) were first estimated using panel least squares and the panel fully-modified least squares (FMOLS) estimation method, using data for the period 1996–2018.

Since nearly 50% of the remittances to Latin America go to Mexico, as Table 2 illustrates, we also analyzed the data for individual countries using the autoregressive distributed lag (ARDL-ECM) approach. This was performed only for the economic growth equation, using data for the period 1980–2018. Since there are a lot of missing values on poverty rate, we did not perform a similar analysis for the poverty equation. In order to carry out this analysis, we used the following models:

$$g_t = \mu + \delta t + \beta_1 l_t + \beta_2 k_t + \beta_3 hc_t + \beta_4 wr_{t-1} + \vartheta_t \tag{10}$$

Equation (10) shows the long-run relationships among the dependent and independent variables in our model. In estimating the long-run model outlined by Equation (10), it is now a common practice to distinguish the short-run effects from the long-run effects. For this purpose, Equation (10) should be specified in an error-correction modeling (ECM) format. This method had been used in many recent studies. Such an approach is warranted given that some variables in Equation (10) can be stationary variables, whereas the other variables could be non-stationary. Therefore, following Pesaran et al. (2001) and their method of the Autoregressive Distributed Lag (ARDL) approach to co-integration analysis, we rewrote Equation (10) as an ARDL-ECM model in Equation (11) below:

$$\Delta g_t = \rho_0 + \rho_1 t + \sum_{i=1}^{n} a_i \Delta g_{t-i} + \sum_{i=0}^{n} b_i \Delta l_{t-i} + \sum_{i=0}^{n} c_i \Delta k_{t-i} + \sum_{i=0}^{n} d_i \Delta hc_{t-i} + \sum_{i=0}^{n} e_i \Delta wr_{t-1-i}$$
$$+ \pi_0 g_{t-1} + \pi_1 l_{t-1} + \pi_2 k_{t-1} + \pi_3 hc_{t-1} + \pi_4 wr_{t-2} + \epsilon_t \tag{11}$$

where $\Delta$ is the difference operator and the other variables are as defined earlier. In addition, n is the lag length and $\epsilon_t$ is a random error term. Pesaran et al. (2001) bounds testing approach to co-integration is based on two procedural steps. The first step involves using an F-test for joint significance of the no co-integration hypothesis $H_o: \pi_0 = \pi_1 = \pi_2 = \pi_3 = \pi_4 = 0$. After establishing co-integration, the second step involves estimation of the following error-correction model to examine short-run effects.

$$\Delta g_t = \alpha_0 + \alpha_1 t + \beta_0 \vartheta_{t-1} + \sum_{i=1}^{k} \beta_i \Delta g_{t-i} + \sum_{i=0}^{k} \gamma_i \Delta l_{t-i} + \sum_{i=0}^{k} \delta_i \Delta k_{t-i} + \sum_{i=0}^{k} \sigma_i \Delta hc_{t-i} + \sum_{i=0}^{k} \theta_i \Delta wr_{t-1-i} + \delta_t \tag{12}$$

where $\vartheta_{t-1}$ is the lagged residual of the co-integration relationship from the model in Equation (10), and $\delta_t$ is a white-noise disturbance term. The lag length k was initially set to 4 lags but insignificant coefficients were successively dropped until the best fit model was found.

### 3.2. Variable Description and Data Sources

In order to test the implications of our models, we collected a panel of aggregate data on remittances and poverty on 21 Latin American countries that covers 39 years (from 1980 to 2018). Though most of the information related to variables included in Equations (8) and (9) are available for 1980–2018, information on poverty is only available until 2017. For this reason, the poverty model given in Equation (9) was estimated using data for the period 1980–2017. Due to unavailability of data on poverty for the entire study period, poverty model in Equation (9) was not estimated at the individual country level. The description of the variables and data sources is provided in Appendix A.

## 4. Empirical Results and Discussion

We have carried out the analyses in two steps. First, the growth model and the poverty model presented in Equations (8) and (9) were estimated using the panel least squares method and panel fully-modified least squares (FMOLS) method. The reason for the use of panel fully-modified least squares (FMOLS) method is to check whether the results obtained from panel least square method still holds when the estimation method is changed. Panel least squares involves estimating a regression model in which the data structure is panel data. In this estimation method, parameter estimation in the regression analysis with panel data is done by estimating the least squares method called the Ordinary Least Squares (OLS) in which the error sum of squares is minimized. The FMOLS estimator developed by Pedroni (2000) produces an asymptotically unbiased estimator of the long-run elasticities and efficient, normally distributed standard errors, if the variables are shown to be cointegrated. In addition, the FMOLS uses a semi-parametric correction for endogeneity and residual autocorrelation and requires fewer assumptions and tends to be more robust. Furthermore, the FMOLS estimator is a group mean or between-group estimator that allows for a high degree of heterogeneity in the panel. Table 3 presents the estimated results of growth that Equation (8) obtained using the panel least squares method while Table 4 presents the estimated results of growth equation (8) obtained using the panel

fully-modified least squares method. Based on the Hausman test, we conclude that the fixed-effects model is the most appropriate estimation method. In each model, we have included several institutional variables, one model with interactive term of remittances with institutional variables. The coefficients of interactive terms can be interpreted as the marginal increase in the impact of remittances on growth when institutional quality improves.

**Table 3.** Worker Remittances and Economic Growth: Panel LS Estimations.

| Variable | Dependent Variable: GDP per Capita Growth ($g_t$) | | | | | | |
|---|---|---|---|---|---|---|---|
| | **Model 1** | **Model 2** | **Model 3** | **Model 4** | **Model 5** | **Model 6** | **Model 7** |
| *Constant* | 0.2577 *** (0.007) | 0.2465 * (0.078) | 0.2788 ** (0.046) | 0.3644 ** (0.011) | 0.2856 ** (0.045) | 0.3497 ** (0.012) | 0.3577 ** (0.011) |
| $y_{t-1}$ | −0.1068 *** (0.000) | −0.1088 *** (0.000) | −0.1218 *** (0.000) | −0.1243 *** (0.000) | −0.139 *** (0.000) | −0.1369 *** (0.000) | −0.1245 *** (0.000) |
| $k_t$ | 0.0644 *** (0.000) | 0.0688 *** (0.000) | 0.0607 *** (0.000) | 0.0623 *** (0.000) | 0.0629 *** (0.000) | 0.0607 *** (0.000) | 0.0623 *** (0.000) |
| $l_t$ | 0.0493 ** (0.018) | 0.0518 ** (0.012) | 0.0630 *** (0.003) | 0.0534 ** (0.010) | 0.0521 ** (0.013) | 0.0684 *** (0.001) | 0.0533 *** (0.000) |
| $hc_t$ | −0.0099 (0.895) | 0.0039 (0.493) | 0.0012 (0.832) | 0.0029 (0.614) | 0.0054 (0.949) | 0.0037 (0.501) | 0.0089 (0.873) |
| $wr_{t-1}$ | 0.0125 *** (0.000) | 0.0119 *** (0.000) | 0.0144 *** (0.000) | 0.0150 *** (0.000) | 0.0146 *** (0.000) | 0.0169 *** (0.000) | 0.0150 *** (0.000) |
| $oda_t$ | 0.0084 *** (0.002) | 0.0080 *** (0.004) | 0.0086 *** (0.002) | 0.0107 *** (0.001) | 0.0081 *** (0.005) | 0.0075 *** (0.007) | 0.0082 *** (0.003) |
| *COCOR* | | 0.0354 *** (0.000) | | | | | |
| *GOVEF* | | | 0.0196 (0.122) | | | | |
| *POLST* | | | | 0.0217 ** (0.015) | | | |
| *REGQU* | | | | | 0.0119 (0.122) | | |
| *RULAW* | | | | | | 0.0296 *** (0.002) | |
| *VOACC* | | | | | | | 0.0351 *** (0.001) |
| *COCOR* x *wr* | | 0.0132 (0.902) | | | | | |
| *GOVEF* x *wr* | | | 0.1936 (0.115) | | | | |
| *POLST* x *wr* | | | | 0.0631 (0.573) | | | |
| *REGQU* x *wr* | | | | | 0.0177 (0.883) | | |
| *RULAW* x *wr* | | | | | | 0.2003 ** (0.021) | |
| *VOACC* x *wr* | | | | | | | 0.0223 (0.893) |
| Number of Observations | 428 | 428 | 428 | 428 | 428 | 428 | 428 |
| Adjusted $R^2$ | 0.263 | 0.290 | 0.280 | 0.281 | 0.267 | 0.294 | 0.289 |
| Random/Fixed Effects? | FE | FE | FE | FE | FE | FE | FE |
| Hausman Test | 54.843 *** | 54.335 *** | 58.023 *** | 61.215 *** | 60.031 *** | 63.960 *** | 60.448 *** |

Note: Institutional variables COCOR represents Control of Corruption, GOVEF represents Government Effectiveness, POLST represents Political Stability and Absence of Violence, REGQU represents Regulatory Quality, RULAW represents Rule of Law, and VOACC represents Voice and Accountability. Figures in parentheses are p-values. Asterisks *, ** and *** indicate the statistical significance at the 10% level, 5% level and 1% level, respectively. Oda is the log of overseas development assistance as a share of GDP.

**Table 4.** Worker Remittances and Economic Growth: Panel FMOLS Estimations.

| Variable | Dependent Variable: GDP per Capita Growth ($g_t$) | | | | | | |
|---|---|---|---|---|---|---|---|
| | **Model 1** | **Model 2** | **Model 3** | **Model 4** | **Model 5** | **Model 6** | **Model 7** |
| $y_{t-1}$ | −0.1079 *** (0.000) | −0.1157 *** (0.000) | −0.1294 *** (0.000) | −0.1299 *** (0.000) | −0.186 *** (0.000) | −0.1445 *** (0.000) | −0.1335 *** (0.000) |
| $k_t$ | 0.0566 *** (0.000) | 0.0643 *** (0.000) | 0.0553 *** (0.000) | 0.0568 *** (0.000) | 0.0588 *** (0.000) | 0.0556 *** (0.000) | 0.0576 *** (0.000) |
| $l_t$ | 0.0403 *** (0.001) | 0.0463 *** (0.000) | 0.0576 *** (0.000) | 0.0487 *** (0.000) | 0.0440 *** (0.000) | 0.0678 *** (0.000) | 0.0491 *** (0.000) |
| $hc_t$ | 0.0010 (0.756) | 0.0073 ** (0.014) | 0.0038 (0.183) | 0.055 * (0.054) | 0.0026 (0.342) | 0.0069 ** (0.017) | 0.0041 (0.125) |
| $wr_{t-1}$ | 0.0162 *** (0.000) | 0.0157 *** (0.000) | 0.0184 *** (0.000) | 0.0187 *** (0.000) | 0.0182 *** (0.000) | 0.0206 *** (0.000) | 0.0195 *** (0.000) |
| $oda_t$ | 0.0065 *** (0.001) | 0.0056 *** (0.000) | 0.0062 *** (0.000) | 0.0091 *** (0.001) | 0.0064 *** (0.000) | 0.0057 *** (0.001) | 0.0061 *** (0.001) |
| COCOR | | 0.0374 *** (0.000) | | | | | |
| GOVEF | | | 0.0225 *** (0.000) | | | | |
| POLST | | | | 0.0209 *** (0.000) | | | |
| REGQU | | | | | 0.0079 ** (0.038) | | |
| RULAW | | | | | | 0.0305 *** (0.000) | |
| VOACC | | | | | | | 0.0451 *** (0.000) |
| COCOR × wr | | 0.0624 (0.268) | | | | | |
| GOVEF × wr | | | 0.2346 *** (0.008) | | | | |
| POLST × wr | | | | 0.0903 (0.103) | | | |
| REGQU × wr | | | | | 0.0797 (0.171) | | |
| RULAW × wr | | | | | | 0.2435 *** (0.000) | |
| VOACC × wr | | | | | | | 0.0195 (0.094) |
| Number of Observations | 405 | 405 | 405 | 405 | 405 | 405 | 405 |
| Adjusted $R^2$ | 0.292 | 0.301 | 0.316 | 0.314 | 0.290 | 0.330 | 0.334 |

Note: Institutional variables COCOR represents Control of Corruption, GOVEF represents Government Effectiveness, POLST represents Political Stability and Absence of Violence, REGQU represents Regulatory Quality, RULAW represents Rule of Law, and VOACC represents Voice and Accountability. Figures in parentheses are p-values. Asterisks *, ** and *** indicate the statistical significance at the 10% level, 5% level and 1% level, respectively. Oda is the log of overseas development assistance as a share of GDP.

The results presented in Table 3 show that capital stock variable has the expected positive sign and it is highly statistically significant in all estimations of Equation (8). The labor force variable also has the expected positive sign and is highly statistically significant in all cases. The human capital variable also has the expected positive sign in six of the seven models estimated and it is not statistically significant in any of the models. The coefficient of workers' remittances variable has the positive sign

in all cases and it is highly statistically significant in all seven cases. This finding suggests that workers' remittances contribute to economic growth in Latin American countries during the study period. These findings are consistent with the findings of some previous studies, such as Giuliano and Ruiz-Arranz (2009); Mundaca (2009); Nsiah and Fayissa (2013); Pradhan et al. (2008); and Fajnzylber and López (2007). The net overseas development assistance flows variable has a positive impact on economic growth in Latin America. The coefficient of this variable is also statistically significant in all cases. The institutional variables have positive signs in both direct and interactive terms, implying that the institutional environment is vital for enhancing the growth and development impact of remittances in Latin America.

Table 4 also show that capital stock variable has the expected positive sign and it is highly statistically significant in all seven models. The labor force variable also has the expected positive sign and is highly statistically significant in all cases. The human capital variable also has the expected positive sign in all models estimated and it is statistically significant in three of the seven models. The coefficient of workers' remittances variable has the positive sign in all cases and it is highly statistically significant in all seven cases. The net overseas development assistance flows variable has a positive impact on economic growth in Latin America. The coefficient of this variable is also statistically significant in all cases. The institutional variables have positive signs in both direct and interactive terms. The results presented in Table 4 are consistent with the results presented in Table 4, indicating that the results do not depend on the estimation method.

Tables 5 and 6 present the estimated results of the poverty level obtained from Equation (9). Table 5 presents the estimated results of the poverty level obtained from Equation (9) using the panel least squares method, while Table 6 presents the estimated results of growth that Equation (9) obtained using the panel fully-modified least squares method. Since the Hausman test statistic is statistically significant, the poverty equation was also estimated using the fixed-effects model. The results presented in Table 5 show that the real GDP per capita variable has the expected negative sign and it is statistically significant in all cases at the 1 percent level of significance. As the per capita income improves, it is expected to lower the poverty rate. The coefficient of lagged poverty variable has the expected positive and it is statistically significant in all seven models. The worker remittances variable has a negative sign, suggesting that workers' remittances tend to lower the poverty rates in Latin America. However, this variable is not statistically significant in two of the seven models. These findings are consistent with those of some previous studies such as Vacaflores (2018); Akobeng (2016); Adams and Page (2003, 2005); Acosta et al. (2008); and Adams and Cuecuecha (2013). The overseas development assistance variable has a negative but statistically insignificant effect on the poverty rate. This suggests that, similar to worker remittances, foreign aid also tend to lower poverty rates in Latin America. The institutional variables have negative signs in all models except for one. The higher the level of remittances to Latin America, the lower the level of poverty in these nations. The coefficient of institutional variable is statistically significant only for interactive terms in four models. This finding suggests that the institutional environment in combination with increased workers' remittances is important for alleviating poverty in Latin America. Therefore, allowing a higher degree of mobility into developed nations can be an efficient instrument for reducing poverty.

The results presented in Table 6 show that the real GDP per capita variable has the expected negative sign and it is highly statistically significant in all cases. The coefficient of lagged poverty variable has the expected positive and it is statistically significant in all seven models. The worker remittances variable has a negative sign, suggesting that workers' remittances tend to lower the poverty rates in Latin America. In this case, the workers' remittances variable is statistically significant in all seven models. The overseas development assistance variable has a negative but statistically insignificant effect on poverty rate. The institutional variables have negative signs in all of the models except for two. The coefficient of institutional variable is only statistically significant for interactive terms in four of the models. The results presented in Table 6 are consistent with the results presented

in Table 5, indicating that the results do not depend on the estimation method, as in the case of growth models.

**Table 5.** Worker Remittances and Poverty: Panel LS Estimation.

| Variable | Dependent variable: Poverty Rate ($pov_t$) | | | | | | |
|---|---|---|---|---|---|---|---|
| | **Model 1** | **Model 2** | **Model 3** | **Model 4** | **Model 5** | **Model 6** | **Model 7** |
| *Constant* | 9.5972 *** (0.000) | 9.9882 *** (0.000) | 9.2214 *** (0.000) | 8.2576 *** (0.000) | 9.8806 *** (0.000) | 9.6778 *** (0.000) | 9.4023 *** (0.000) |
| $pov_{t-1}$ | 0.6608 *** (0.000) | 0.6301 *** (0.000) | 0.6105 *** (0.000) | 0.6585 *** (0.000) | 0.6418 *** (0.000) | 0.6571 *** (0.000) | 0.6581 *** (0.000) |
| $y_t$ | −1.1580 *** (0.000) | −1.1902 *** (0.000) | −1.2068 *** (0.000) | −1.1122 *** (0.000) | −1.1882 *** (0.000) | −1.1601 *** (0.000) | −1.1281 *** (0.000) |
| $wr_{t-1}$ | −0.0244 (0.129) | −0.0488 *** (0.009) | −0.0509 *** (0.003) | −0.0360 ** (0.041) | −0.0251 (0.175) | −0.0337 * (0.078) | −0.0333 * (0.052) |
| $oda_t$ | −0.0124 (0.515) | −0.0113 (0.543) | −0.0107 (0.186) | −0.0169 (0.383) | −0.0199 (0.599) | −0.0197 (0.612) | −0.0195 (0.620) |
| COCOR | −0.0424 (0.539) | | | | | | |
| GOVEF | | | −0.0501 (0.561) | | | | |
| POLST | | | | −0.0436 (0.483) | | | |
| REGQU | | | | | −0.0781 (0.134) | | |
| RULAW | | | | | | 0.0631 (0.364) | |
| VOACC | | | | | | | −0.0047 (0.957) |
| COCOR × wr | | −2.7445 *** (0.000) | | | | | |
| GOVEF × wr | | | −2.7957 *** (0.001) | | | | |
| POLST × wr | | | | −0.7454 (0.381) | | | |
| REGQU × wr | | | | | −2.5623 *** (0.000) | | |
| RULAW × wr | | | | | | −1.1572 * (0.064) | |
| VOACC × wr | | | | | | | −1.8652 (0.141) |
| Number of Observations | 358 | 358 | 358 | 358 | 358 | 358 | 358 |
| Adjusted $R^2$ | 0.966 | 0.970 | 0.968 | 0.966 | 0.963 | 0.966 | 0.966 |
| Random/Fixed Effects? | FE | FE | FE | FE | FE | FE | FE |
| Hausman Test | 91.123 *** | 98.604 *** | 99.939 *** | 86.041 *** | 91.032 *** | 95.212 *** | 87.370 *** |

Note: Institutional variables COCOR represents Control of Corruption, GOVEF represents Government Effectiveness, POLST represents Political Stability and Absence of Violence, REGQU represents Regulatory Quality, RULAW represents Rule of Law, and VOACC represents Voice and Accountability. Figures in parentheses are p-values. Asterisks *** and * indicate the statistical significance at the 1% level and 10% level, respectively. Oda is the log of overseas development assistance as a share of GDP.

**Table 6.** Worker Remittances and Poverty: Panel FMOLS Estimation.

| Variable | Dependent Variable: Poverty Rate ($pov_t$) | | | | | | |
|---|---|---|---|---|---|---|---|
| | **Model 1** | **Model 2** | **Model 3** | **Model 4** | **Model 5** | **Model 6** | **Model 7** |
| $pov_{t-1}$ | 0.6697 *** (0.000) | 0.6306 *** (0.000) | 0.6081 *** (0.000) | 0.6618 *** (0.000) | 0.6409 *** (0.000) | 0.6501 *** (0.000) | 0.6619 *** (0.000) |
| $y_t$ | −1.0932 *** (0.000) | −1.1355 *** (0.000) | −1.1666 *** (0.000) | −1.0554 *** (0.000) | −1.1448 *** (0.000) | −1.1320 *** (0.000) | −1.0861 *** (0.000) |
| $wr_{t-1}$ | −0.0274** (0.020) | −0.0548 *** (0.000) | −0.0589 *** (0.000) | −0.0435 *** (0.000) | −0.0367 *** (0.004) | −0.0400 *** (0.002) | −0.0419 *** (0.000) |
| $oda_t$ | −0.0196 (0.122) | −0.0151 (0.215) | −0.0124 (0.268) | −0.0212 * (0.067) | −0.0109 (0.343) | −0.0173 (0.139) | −0.0163 (0.169) |
| COCOR | | −0.0237 (0.611) | | | | | |
| GOVEF | | | −0.0468 (0.389) | | | | |
| POLST | | | | −0.0462 (0.224) | | | |
| REGQU | | | | | −0.0640 * (0.057) | | |
| RULAW | | | | | | 0.0581 (0.191) | |
| VOACC | | | | | | | 0.0069 (0.907) |
| COCOR × wr | | −2.7048 *** (0.000) | | | | | |
| GOVEF × wr | | | −2.7120 *** (0.000) | | | | |
| POLST × wr | | | | −0.3288 (0.529) | | | |
| REGQU × wr | | | | | −2.3143 *** (0.000) | | |
| RULAW × wr | | | | | | −1.0706 *** (0.008) | |
| VOACC × wr | | | | | | | −1.5189 * (0.064) |
| Number of Observations | 333 | 328 | 328 | 328 | 328 | 328 | 328 |
| Adjusted $R^2$ | 0.966 | 0.967 | 0.966 | 0.965 | 0.966 | 0.965 | 0.965 |

Note: Institutional variables COCOR represents Control of Corruption, GOVEF represents Government Effectiveness, POLST represents Political Stability and Absence of Violence, REGQU represents Regulatory Quality, RULAW represents Rule of Law, and VOACC represents Voice and Accountability. Figures in parentheses are *p*-values. Asterisks *** and * indicate the statistical significance at the 1% level and 10% level, respectively. Oda is the log of overseas development assistance as a share of GDP.

In a second step, the growth model was estimated for each of the 21 countries using the autoregressive distributed lag (ARDL-ECM) approach to co-integration analysis in order to estimate the short-run and long-run effects of workers' remittances on economic growth in each country. Unit root tests were conducted using the Augmented Dickey-Fuller (ADF) test and the results are presented in Table A3 in Appendix B.

While there are several tests (such as the Dickey-Fuller GLS test, Phillips-Perron test, Kwiatkowski-Phillips-Schmidt-Shin test, Elliott-Rothenberg-Stock test statistic, and Ng-Perron test) are available to testing for unit roots, we have employed the ADF test since it is the most commonly used test. It is evident from the results that all variables are non-stationary at the levels, but are stationary at

the first difference. The optimal lag length based on Schwarz criterion (SIC) is also reported for each of the test statistics.

Applying the ARDL-ECM approach to co-integration, we assess the co-integrating relationships for each of the 21 Latin American for the growth model given in Equation (8). We imposed a maximum of four lags on each first differenced variable and employed Akaike's Information Criterion (AIC) to select the optimum lag length. Choosing a combination of lags that minimizes the AIC, we then tested whether the variables for each country are co-integrated. The results of the co-integration analysis are presented in Table A4 in Appendix B. This study does not pursue the autoregressive distributed lag approach in a panel context due to missing data on several variables.

Table A4 reveals that 16 of the 21 countries encompass an F-statistic above the upper bound, implying that the four variables are co-integrated in these countries. Therefore, we concluded that either there exists a long-run relationship among the variables, or that the five variables in our models are co-integrated for these sixteen countries. The estimated coefficients for the long-run relationships for these countries are presented in Table 7. Hence, five countries, namely, Belize, Honduras, Peru, Suriname, and Venezuela, were dropped from further analysis and only the estimated coefficients for the long-run relationships for sixteen countries are presented in Table 7. Workers' remittances have a positive effect on long-run economic growth in 15 of the 16 countries. The coefficient of the workers' remittances variable is positive and statistically significant either at the 1 percent or 5 percent level of significance in half of them (Argentina, Chile, Costa Rica, El Salvador, Guatemala, Guyana, Mexico, and Panama). Workers' remittances in Uruguay has a negative and statistically significant effect on economic growth in the long-run.

The results of the error-correction model for each of the sixteen countries are presented in Appendix B Table A5. The error-correction term is highly statistically significant in all cases. The short-run estimated coefficients on workers' remittances variables reveal a mixture of negative and positive signs while the majority of them have positive signs. The coefficient of the workers' remittances variable is positive and statistically significant either at the 1 percent or 5 percent level of significance in the Argentina, Bolivia, Brazil, Chile, and Dominican Republic in the short-run. In the rest of the countries, workers' remittances tend to have a mix of negative and positive effects on economic growth in the short-run. Since none of the previous studies has used the ARDL models to estimate short-run effects of workers' remittances, it is not possible for us to compare our results with other studies. It should be noted that we did not include institutional variables when we estimated the ARDL models. The reason for not including these variables is that differences in institutional settings are relevant only when a group of countries are included in the analysis.

**Table 7.** Long-run Relationship Estimates.

| Country | Constant | $l_t$ | $k_t$ | $hc_t$ | $wr_{t-1}$ | $oda_t$ |
|---|---|---|---|---|---|---|
| Argentina | 0.020 (0.967) | 1.077 *** (0.000) | 0.453 *** (0.000) | 0.272 ** (0.012) | 0.032 ** (0.033) | −0.003 (0.791) |
| Bolivia | −2.898 *** (0.000) | 2.182 *** (0.000) | 0.119* (0.078) | −0.297 *** (0.003) | 0.010 (0.630) | −0.208 *** (0.000) |
| Brazil | 1.064 * (0.053) | 0.970 *** (0.000) | 0.126 ** (0.012) | 0.114 ** (0.024) | 0.024 (0.225) | 0.064 *** (0.000) |
| Chile | −1.230** (0.055) | 1.640 *** (0.000) | 0.062 (0.584) | 0.120 * (0.059) | 0.044 * (0.066) | 0.009 (0.531) |
| Colombia | 1.282 *** (0.002) | 0.936 *** (0.001) | 0.069 (0.488) | −0.255 *** (0.003) | 0.024 (0.430) | 0.054 * (0.075) |
| Costa Rica | 0.886 (0.132) | 1.398 *** (0.000) | 0.761 *** (0.000) | −0.253 *** (0.003) | 0.069 ** (0.042) | −0.253 *** (0.005) |
| Dominican Republic | −1.408 (0.116) | 1.535 *** (0.000) | 0.063 (0.652) | 0.026 (0.679) | 0.156** (0.024) | −0.032 (0.153) |

**Table 7.** *Cont.*

| Country | Constant | $l_t$ | $k_t$ | $hc_t$ | $wr_{t-1}$ | $oda_t$ |
|---|---|---|---|---|---|---|
| Ecuador | −0.554 | 1.366 *** | 0.379 *** | −0.167 | 0.092 | −0.026 |
| | (0.246) | (0.000) | (0.008) | (0.481) | (0.526) | (0.628) |
| El Salvador | −0.324 | 1.418 *** | 0.573 | 0.012 | 0.237 ** | −0.085 *** |
| | (0.617) | (0.001) | (0.132) | (0.724) | (0.021) | (0.000) |
| Guatemala | 0.091 | 1.206 *** | 0.068 | −0.007 | 0.012 * | −0.038 * |
| | (0.636) | (0.000) | (0.144) | (0.825) | (0.065) | (0.076) |
| Guyana | 0.231 | 0.855 | 0.225 | 0.333 ** | 0.203 *** | 0.235 ** |
| | (0.935) | (0.462) | (0.197) | (0.019) | (0.000) | (0.043) |
| Mexico | 1.833 *** | 0.847 *** | 0.011 | −0.190 * | 0.105 ** | −0.010 |
| | (0.000) | (0.000) | (0.926) | (0.069) | (0.033) | (0.358) |
| Nicaragua | 2.490* | 0.403 | 0.245 | 0.195 | 0.099 | −0.236 *** |
| | (0.071) | (0.434) | (0.171) | (0.780) | (0.105) | (0.000) |
| Panama | −2.285 *** | 2.269 *** | 0.267 *** | −1.284 *** | 0.184 *** | 0.010 |
| | (0.000) | (0.000) | (0.000) | (0.000) | (0.000) | (0.657) |
| Paraguay | 0.308 | 1.137 *** | 0.031 | −0.126 * | 0.023 | 0.138 *** |
| | (0.565) | (0.000) | (0.864) | (0.053) | (0.536) | (0.002) |
| Uruguay | −1.398 *** | 1.972 *** | 0.257 ** | −0.147 *** | −0.155 *** | 0.035 ** |
| | (0.000) | (0.000) | (0.010) | (0.009) | (0.003) | (0.017) |

Notes: This table summarizes the results of the long-run relationship estimates. The figures in parentheses are *p*-values. * 10 percent level, ** and *** indicate the statistical significance at the 5 and 1 percent level of significance, respectively.

## 5. Conclusions

This paper analyzes the effects of workers' remittances on the economic growth and poverty in 21 Latin American countries. We used the panel least square method and panel fully-modified least squares (FMOLS) methods for all countries as well as the autoregressive distributed lag (ARDL-ECM) approach to co-integration analysis to estimate the short-run and long-run effects of workers' remittances on economic growth for each individual Latin American country. Since there are a lot of missing values in poverty rates for Latin American countries, we have not performed a co-integration analysis to estimate the short-run and long-run effects of workers' remittances on poverty for individual countries.

Since the Hausman test statistic is statistically significant, the economic growth equation outlined in Equation (8) was estimated using the fixed-effects model. When the economic growth model was estimated using the panel data covering all countries it was found that workers' remittances variable has a positive sign and it is highly statistically significant. The results were somewhat similar regardless of whether the models were estimated using the fixed-effects OLS method or the FMOLS method. This finding suggests that workers' remittances contributed positively to economic growth in Latin American countries during the study period. As in the case of the estimation of the growth model, since the Hausman test statistic is statistically significant, the poverty equation outlined in Equation (9) was also estimated using the fixed-effects model. The results of the estimated model suggest that the remittances variable has a negative sign, suggesting that workers' remittances tend to lower the poverty rates in Latin America. In addition, this variable was generally found to be highly statistically significant.

Our method of bounds test revealed that sixteen of the 21 countries encompass an F-statistic above the upper bound, implying that the five variables are co-integrated in these countries. Workers' remittances have a positive effect on long-run economic growth in 15 of the 16 countries. The coefficient of the workers' remittances variable was found to be positive and statistically significant in eight of them. Workers' remittances have a negative and statistically significant effect on economic growth in

the long-run only for Uruguay. The short-run estimated coefficients on workers' remittances variables were also found to be mostly positive, though some coefficients were not statistically significant.

In developing countries, remittances play an important role as a stable source of household income. Remittances can also improve the credit constraints on the poor, improve the allocation of capital, and substitute for the lack of financial development. Financial sector can be a channel though which remittances affect economic growth. Remittances are used to raise national savings, reduce the constraint associated with foreign exchange and balance of payments, and contribute to development budget. Remittances also increase the rate of accumulation of both physical and human capital, in addition to lowering the cost of capital in the recipient country. Another channel through which remittances contribute to economic growth is their positive impact on consumption, savings, investment, and entrepreneurship. The quality of the receiving country's policies and institutions also enhance the growth effects of remittances. Many of these channels work through investment in physical and human capital. In this study, since physical capital, human capital, and governance/institutional variables are already controlled for in the regressions, we acknowledge that the identification of additional channels is an important topic for further research.

**Author Contributions:** Each author contributed equally to the research article. Data analysis was undertaken mainly by the first author. All authors have read and agreed to the published version of the manuscript.

**Funding:** This research received no external funding.

**Conflicts of Interest:** The authors declare no conflict of interest.

## Appendix A Data Definitions and Sources

*Economic Growth:* The economic growth rate is measured in this study as the growth of real per capita GDP in constant (2010) U.S. dollars. The data on real GDP per capita are from the World Bank (2019b), *World Development Indicators 2019* database.

*Labor Stock:* Labor stock is measured by the total labor force, people aged 15 and older. The data on labor force are from the United Nations Conference on United Nations Conference on Trade and Development (UNCTAD), *UNCTADSTAT* database.

*Capital Stock:* The investment/GDP ratio is used as a proxy for the growth rate of the capital stock. Since the investment/GDP ratio is not reported for the majority of the developing countries, gross fixed capital formation as a share of GDP is used to represent investment/GDP ratio. The data on investment/GDP ratio are also from the World Bank (2019b), *World Development Indicators 2019* database.

*Human Capital:* The share of population by educational attainment (Bachelor's or equivalent), population 25 years and older is used as a measure of human capital. The data on educational attainment are from the 2019) United Nations Educational and (UNESCO), *UNESCO Institute for Statistics (UIS) 2019* database.

*Remittances:* The data on worker remittances are from the United Nations Conference on United Nations Conference on Trade and Development (UNCTAD), *UNCTADSTAT* database and from the World Bank (2019b), *World Development Indicators 2019* database. According to the World Bank (2019b), personal remittances comprise personal transfers and compensation of employees. Personal transfers consist of all current transfers in cash or in kind made or received by resident households to or from nonresident households. Personal transfers thus include all current transfers between resident and nonresident individuals. Compensation of employees refers to the income of border, seasonal, and other short-term workers who are employed in an economy where they are not resident and of residents employed by nonresident entities. Data are the sum of two items, namely, personal transfers and compensation of employees.

*Foreign Aid:* The data on net official development assistance flows are from the International Monetary Fund (IMF), *Balance of Payments Yearbook 2019* database. Net official development assistance (ODA) consists of disbursements of loans made on concessional terms (net of repayments of principal) and grants by official agencies of the members of the Development Assistance Committee (DAC), by

multilateral institutions, and by non-DAC countries to promote economic development and welfare in countries and territories in the DAC list of ODA recipients.

*Poverty Rate:* We have used the poverty headcount ratio at $1.90 a day as our measure of poverty. It is the percentage of the population living on less than $1.90 a day at 2011 international prices. The data on poverty headcount ratio are from the World Bank (2019b), *World Development Indicators 2019* database.

*Control of Corruption:* Reflects perceptions of the extent to which public power is exercised for private gain, including both petty and grand forms of corruption, as well as "capture" of the state by elites and private interests. Estimate of governance (ranges from approximately −2.5 (weak) to 2.5 (strong) governance performance). The data on control of corruption are from the World Bank (2018), *The Worldwide Governance Indicators, 2018* Update database.

*Government Effectiveness:* Reflects perceptions of the quality of public services, the quality of the civil service and the degree of its independence from political pressures, the quality of policy formulation and implementation, and the credibility of the government's commitment to such policies. Estimate of governance (ranges from approximately −2.5 (weak) to 2.5 (strong) governance performance). The data on government effectiveness are from the World Bank (2018), *The Worldwide Governance Indicators, 2018 Update* database.

*Political Stability and Absence of Violence/Terrorism:* Political Stability and Absence of Violence/Terrorism measures perceptions of the likelihood of political instability and/or politically-motivated violence, including terrorism. Estimate of governance (ranges from approximately −2.5 (weak) to 2.5 (strong) governance performance). The data on political stability and absence of violence/terrorism are from the World Bank (2018), *The Worldwide Governance Indicators, 2018 Update* database.

*Regulatory Quality:* Reflects perceptions of the ability of the government to formulate and implement sound policies and regulations that permit and promote private sector development. Estimate of governance (ranges from approximately −2.5 (weak) to 2.5 (strong) governance performance). The data on regulatory quality are from the World Bank (2018), *The Worldwide Governance Indicators, 2018 Update* database.

*Rule of Law:* Reflects perceptions of the extent to which agents have confidence in and abide by the rules of society, and in particular the quality of contract enforcement, property rights, the police, and the courts, as well as the likelihood of crime and violence. Estimate of governance (ranges from approximately −2.5 (weak) to 2.5 (strong) governance performance). The data on rule of law are from the World Bank (2018), *The Worldwide Governance Indicators, 2018 Update* database.

*Voice and Accountability:* Reflects perceptions of the extent to which a country's citizens are able to participate in selecting their government, as well as freedom of expression, freedom of association, and a free media. Estimate of governance (ranges from approximately −2.5 (weak) to 2.5 (strong) governance performance). The data on voice and accountability are from the World Bank (2018), *The Worldwide Governance Indicators, 2018 Update* database.

## Appendix B

**Table A1.** Gross Domestic Product (GDP) in Latin America, 1980–2018.

| Country | Gross Domestic Product (US$ Billions) | | | | | | | | 1980–2018 Annual Avg. Growth (%) |
|---|---|---|---|---|---|---|---|---|---|
| | 1980 | 1985 | 1990 | 1995 | 2000 | 2005 | 2010 | 2018 | |
| Argentina | 77.0 | 88.4 | 141.4 | 258.0 | 284.2 | 198.7 | 423.6 | 519.9 | 2.01 |
| Belize | 0.2 | 0.2 | 0.4 | 0.6 | 0.8 | 1.1 | 1.4 | 1.9 | 4.68 |
| Bolivia | 4.5 | 5.4 | 4.9 | 6.7 | 8.4 | 9.5 | 19.6 | 40.3 | 2.99 |
| Brazil | 235.0 | 222.9 | 462.0 | 769.3 | 655.4 | 891.6 | 2208.9 | 1868.6 | 2.43 |
| Chile | 29.0 | 17.7 | 33.1 | 73.4 | 77.9 | 123.0 | 218.5 | 298.2 | 4.39 |
| Colombia | 33.4 | 34.9 | 47.8 | 92.5 | 99.9 | 145.2 | 286.1 | 331.0 | 3.48 |
| Costa Rica | 4.8 | 3.9 | 5.7 | 11.5 | 14.9 | 19.9 | 37.3 | 60.1 | 3.76 |

**Table A1.** *Cont.*

| Country | Gross Domestic Product (US$ Billions) | | | | | | | | 1980–2018 Annual Avg. Growth (%) |
|---|---|---|---|---|---|---|---|---|---|
| | 1980 | 1985 | 1990 | 1995 | 2000 | 2005 | 2010 | 2018 | |
| Dom. Rep. | 6.8 | 5.0 | 7.1 | 16.6 | 24.3 | 36.1 | 53.9 | 85.6 | 4.67 |
| Ecuador | 17.9 | 17.1 | 15.2 | 24.4 | 18.3 | 41.5 | 69.6 | 108.4 | 2.99 |
| El Salvador | 3.6 | 3.8 | 4.8 | 8.9 | 11.8 | 14.7 | 18.4 | 26.1 | 1.43 |
| Guatemala | 7.9 | 9.7 | 7.7 | 14.7 | 19.3 | 27.2 | 41.3 | 78.5 | 2.97 |
| Guyana | 0.6 | 0.5 | 0.4 | 0.6 | 0.7 | 0.8 | 2.3 | 3.9 | 1.92 |
| Honduras | 4.0 | 5.3 | 4.9 | 5.3 | 7.1 | 9.7 | 15.8 | 24.0 | 3.42 |
| Mexico | 205.1 | 195.2 | 261.3 | 360.1 | 707.9 | 877.5 | 1057.8 | 1220.7 | 2.57 |
| Nicaragua | 2.2 | 2.7 | 1.0 | 4.1 | 5.1 | 6.3 | 8.8 | 13.1 | 2.28 |
| Panama | 4.6 | 6.5 | 6.4 | 9.6 | 12.3 | 16.4 | 29.4 | 65.1 | 4.95 |
| Paraguay | 4.4 | 3.3 | 5.8 | 9.1 | 8.9 | 10.7 | 27.2 | 40.5 | 3.79 |
| Peru | 18.1 | 16.5 | 26.4 | 53.3 | 51.7 | 76.1 | 147.5 | 222.0 | 3.31 |
| Suriname | 0.8 | 0.9 | 0.4 | 0.7 | 0.9 | 1.8 | 4.4 | 3.6 | 1.48 |
| Uruguay | 10.2 | 4.7 | 9.3 | 19.3 | 22.8 | 17.4 | 40.3 | 59.6 | 2.47 |
| Venezuela | 59.1 | 62.0 | 48.6 | 77.4 | 117.1 | 145.5 | 393.2 | 482.4 | 0.26 |
| Latin America | 729.2 | 706.8 | 1094.6 | 1816.3 | 2149.9 | 2670.8 | 5105.3 | 5553.4 | 2.45 |

Source: World Bank, *World Development Indicators 2019* Database.

**Table A2.** Poverty Rates in Latin America, 1980–2017.

| Country | Poverty Rate [Poverty Headcount Ratio at $1.90 a Day] (%) | | | | | | | | 1980–2017 Annual Avg. Rate (%) |
|---|---|---|---|---|---|---|---|---|---|
| | 1980 | 1985 | 1990 | 1995 | 2000 | 2005 | 2010 | 2017 | |
| Argentina | 0.4 | | | 4.1 | 5.7 | 3.7 | 2.2 | 0.5 | 3.3 |
| Belize | | | | 11.6 | 13.9 | | | | 12.7 |
| Bolivia | | | 7.1 | | 28.6 | 19.3 | 8.9 | 5.8 | 14.1 |
| Brazil | | 23.1 | 21.6 | 13.0 | 12.5 | 8.6 | 5.1 | 4.8 | 12.9 |
| Chile | | | 8.1 | 4.2 | 4.4 | | 2.1 | 0.7 | 3.6 |
| Colombia | | | | | 16.4 | 9.7 | 7.7 | 3.9 | 10.1 |
| Costa Rica | | | 9.9 | 6.9 | 6.5 | 3.1 | 1.5 | 1.0 | 5.7 |
| Dominican Rep. | | | | | 5.5 | 5.6 | 2.5 | 1.6 | 4.5 |
| Ecuador | | | | | 28.2 | 12.1 | 5.6 | 3.2 | 10.2 |
| El Salvador | | | 19.4 | 12.5 | 12.2 | 10.4 | 5.5 | 1.9 | 10.2 |
| Guatemala | | | | | 9.2 | 11.1 | | 8.8 | 18.6 |
| Guyana | | | 33.9 | | 14.0 | | | | 24.0 |
| Honduras | | | 44.3 | 27.7 | 23.9 | 26.5 | 15.0 | 17.2 | 23.6 |
| Mexico | | | | 15.3 | 9.1 | 6.7 | 4.6 | 2.2 | 8.0 |
| Nicaragua | | | | | | 8.3 | | | 16.0 |
| Panama | 8.1 | | 22.9 | 16.3 | 12.4 | 10.0 | 4.5 | 2.5 | 10.2 |
| Paraguay | | | 1.2 | 12.4 | 9.6 | 6.1 | 5.5 | 1.2 | 6.5 |
| Peru | | | | | 16.3 | 15.3 | 5.5 | 3.4 | 10.3 |
| Suriname | | | | | 23.4 | | | | 23.4 |
| Uruguay | | | | 0.4 | 0.4 | 0.7 | 0.1 | 0.1 | 0.4 |
| Venezuela | | | | 10.4 | 11.5 | 18.9 | | | 12.3 |
| Latin America | 4.3 | 23.1 | 16.8 | 11.2 | 12.5 | 10.3 | 5.1 | 3.4 | 12.0 |

*Source: World Bank,* World Development Indicators 2019 Database.

**Table A3.** Unit Root Tests Statistics.

| | *g* | | *l* | | *k* | | *wr* | | *hc* | |
|---|---|---|---|---|---|---|---|---|---|---|
| **Level** | **ADF** | *p* | **ADF** | *p* | **ADF** | *p* | **ADF** | *p* | **ADF** | *p* |
| Argentina | −1.389 | 1 | −1.765 | 0 | −1.002 | 0 | −0.023 | 0 | −2.154 | 0 |
| Belize | −0.546 | 0 | −0.435 | 1 | −0.721 | 0 | −0.358 | 1 | −1.305 | 0 |
| Bolivia | −2.394 | 0 | −0.628 | 0 | −0.279 | 0 | −0.613 | 2 | −1.643 | 0 |
| Brazil | −1.797 | 0 | −1.919 | 0 | −3.188 | 0 | −1.338 | 0 | −1.442 | 0 |
| Chile | −2.059 | 0 | −0.645 | 0 | −2.509 | 0 | −1.315 | 0 | −0.714 | 2 |
| Colombia | −0.850 | 1 | −2.507 | 1 | −2.874 | 1 | −0.061 | 0 | −0.844 | 1 |
| Costa Rica | −1.614 | 2 | −2.197 | 0 | −3.325* | 0 | −0.636 | 0 | −2.947 * | 4 |
| Dominican Rep. | −2.735 * | 0 | −0.533 | 0 | −2.864 | 0 | −2.718 | 0 | −1.610 | 0 |
| Ecuador | −1.462 | 0 | −2.619 | 0 | −2.192 | 0 | −0.911 | 1 | −1.227 | 0 |
| El Salvador | −0.553 | 1 | −1.497 | 0 | −2.023 | 0 | −0.113 | 1 | −0.668 | 0 |
| Guatemala | −0.286 | 2 | −0.378 | 1 | −1.860 | 0 | −0.867 | 1 | −1.253 | 0 |
| Guyana | −0.630 | 1 | −0.494 | 1 | −0.935 | 0 | −3.538* | 5 | −1.219 | 0 |
| Honduras | −0.880 | 0 | −1.295 | 1 | −0.500 | 0 | −1.478 | 1 | −1.811 | 1 |
| Mexico | −0.281 | 0 | −2.746 | 0 | −0.841 | 0 | −2.153 | 1 | −1.583 | 0 |
| Nicaragua | −1.372 | 1 | −0.649 | 0 | −0.386 | 0 | −0.752 | 0 | −2.034 | 0 |
| Panama | −1.192 | 2 | −2.890 | 0 | −3.163* | 5 | −0.399 | 0 | −2.118 | 0 |
| Paraguay | −0.923 | 0 | −0.810 | 0 | −0.339 | 0 | −1.804 | 0 | −0.379 | 1 |
| Peru | −0.339 | 0 | −2.035 | 0 | −2.424 | 0 | −2.184 | 1 | −0.254 | 0 |
| Suriname | −0.277 | 3 | −2.056 | 0 | −2.685 | 0 | −1.804 | 0 | −2.040 | 0 |
| Uruguay | −1.826 | 1 | −2.024 | 0 | −2.197 | 0 | −3.542* | 1 | −1.721 | 0 |
| Venezuela | −2.129 | 0 | −2.716 | 1 | −0.703 | 0 | −1.815 | 8 | −3.125 * | 0 |
| | Δ*g* | | Δ*l* | | Δ*k* | | Δ*wr* | | Δ*hc* | |
| **First Difference** | **ADF** | *p* | **ADF** | *p* | **ADF** | *p* | **ADF** | *p* | **ADF** | *p* |
| Argentina | −4.094 ** | 0 | −5.004 *** | 0 | −3.130 *** | 8 | −3.929 *** | 0 | −3.518 *** | 1 |
| Belize | −5.347 *** | 0 | −4.765 *** | 0 | −6.859 *** | 0 | −3.523 ** | 0 | −4.272 *** | 1 |
| Bolivia | −3.317 * | 0 | −5.159 *** | 0 | −6.383 *** | 0 | −3.726 *** | 1 | −6.240 *** | 0 |
| Brazil | −4.778 *** | 4 | −4.997 *** | 0 | −4.368 *** | 0 | −5.726 *** | 0 | −5.354 *** | 0 |
| Chile | −3.691 *** | 0 | −4.954 *** | 0 | −6.109 *** | 0 | −5.363 *** | 0 | −3.826 *** | 1 |
| Colombia | −3.118 ** | 0 | −3.318 * | 0 | −4.495 *** | 0 | −4.379 *** | 0 | −7.476 *** | 0 |
| Costa Rica | −5.552 *** | 1 | −7.218 *** | 0 | −6.406 *** | 0 | −4.041 *** | 0 | −8.815 *** | 0 |
| Dominican Rep. | −3.518 ** | 0 | −4.535 *** | 0 | −5.153 *** | 0 | −4.650 *** | 0 | −7.683 *** | 0 |
| Ecuador | −5.791 *** | 0 | −6.185 *** | 0 | −4.267 ** | 1 | −2.777 * | 0 | −6.796 *** | 0 |
| El Salvador | −4.391 *** | 0 | −5.123 *** | 0 | −4.044 ** | 3 | −3.274 ** | 0 | −5.173 *** | 0 |
| Guatemala | −3.812 *** | 1 | −3.261 * | 1 | −4.676 *** | 1 | −3.090 ** | 0 | −5.524 *** | 0 |
| Guyana | −3.209 ** | 1 | −3.415 * | 0 | −5.780 *** | 1 | −3.384 ** | 8 | −6.294 *** | 1 |
| Honduras | −4.583 *** | 0 | −4.222 *** | 0 | −5.742 *** | 0 | −3.608 ** | 1 | −8.911 *** | 0 |
| Mexico | −3.547 ** | 3 | −6.785 *** | 0 | −5.593 *** | 0 | −3.193 ** | 0 | −6.114 *** | 0 |
| Nicaragua | −3.242 ** | 0 | −5.530 *** | 0 | −5.712 *** | 1 | −4.827 *** | 0 | −5.486 *** | 0 |
| Panama | −4.081 ** | 1 | −6.020 *** | 1 | −2.689 *** | 5 | −4.580 *** | 0 | −5.105 *** | 0 |
| Paraguay | −6.591 *** | 0 | −4.405 *** | 0 | −5.961 *** | 0 | −5.542 *** | 0 | −7.452 *** | 0 |
| Peru | −4.620 *** | 0 | −4.825 *** | 0 | −6.384 *** | 0 | −3.620 ** | 0 | −7.114 *** | 0 |
| Suriname | −3.671 ** | 3 | −4.901 *** | 0 | −6.522 *** | 0 | −5.542 *** | 0 | −6.455 *** | 1 |
| Uruguay | −3.609 ** | 0 | −4.777 *** | 0 | −5.649 *** | 0 | −5.895 *** | 1 | −6.256 *** | 0 |
| Venezuela | −4.544 *** | 0 | −3.511 ** | 0 | −5.532 *** | 1 | −3.683 ** | 5 | −6.007 *** | 1 |

Notes: ADF represents the Augmented Dickey-Fuller test statistic and p represents the optimal lag length based on Schwarz criterion (SIC). *, **, and *** indicate the statistical significance at the 10, 5, and 1 percent level of significance, respectively.

**Table A4.** Co-integration Test Results.

| Country | Lags | F-Statistic | Cointegrated? | Critical Values: 10%5%1% |
|---|---|---|---|---|
| Argentina | (4, 4, 3, 4, 2, 4) | 9.449 *** | Yes | I(0) 2.41 2.91 4.13<br>I(1) 3.52 4.19 5.76 |
| Belize | (4, 4, 4, 4, 4, 4) | 2.533 | No | I(0) 2.41 2.91 4.13<br>I(1) 3.52 4.19 5.76 |
| Bolivia | (4, 4, 4, 4, 4, 4) | 9.219 *** | Yes | I(0) 2.41 2.91 4.13<br>I(1) 3.52 4.19 5.76 |
| Brazil | (4, 4, 3, 4, 4, 4) | 7.312 *** | Yes | I(0) 2.41 2.91 4.13<br>I(1) 3.52 4.19 5.76 |
| Chile | (4, 3, 4, 4, 4, 4) | 6.118 *** | Yes | I(0) 2.41 2.91 4.13<br>I(1) 3.52 4.19 5.76 |
| Colombia | (2, 4, 4, 4, 4, 4) | 5.303** | Yes | I(0) 2.41 2.91 4.13<br>I(1) 3.52 4.19 5.76 |
| Costa Rica | (1, 4, 2, 2, 4, 3) | 9.998 *** | Yes | I(0) 2.41 2.91 4.13<br>I(1) 3.52 4.19 5.76 |
| Dominican Rep. | (3, 4, 4, 4, 3, 4) | 6.355 *** | Yes | I(0) 2.41 2.91 4.13<br>I(1) 3.52 4.19 5.76 |
| Ecuador | (4, 4, 4, 4, 3, 4) | 4.571 ** | Yes | I(0) 2.41 2.91 4.13<br>I(1) 3.52 4.19 5.76 |
| El Salvador | (3, 4, 3, 4, 4, 4) | 5.571 *** | Yes | I(0) 2.41 2.91 4.13<br>I(1) 3.52 4.19 5.76 |
| Guatemala | (1, 4, 4, 4, 4, 4) | 7.244 *** | Yes | I(0) 2.41 2.91 4.13<br>I(1) 3.52 4.19 5.76 |
| Guyana | (3, 3, 3, 3, 3, 4) | 5.778 *** | Yes | I(0) 2.41 2.91 4.13<br>I(1) 3.52 4.19 5.76 |
| Honduras | (4, 4, 3, 4, 2, 4) | 2.108 | No | I(0) 2.41 2.91 4.13<br>I(1) 3.52 4.19 5.76 |
| Mexico | (4, 4, 3, 2, 4, 4) | 4.216 ** | Yes | I(0) 2.41 2.91 4.13<br>I(1) 3.52 4.19 5.76 |
| Nicaragua | (4, 4, 4, 3, 4, 4) | 9.318 *** | Yes | I(0) 2.41 2.91 4.13<br>I(1) 3.52 4.19 5.76 |
| Panama | (4, 4, 4, 4, 4, 4) | 4.754 ** | Yes | I(0) 2.41 2.91 4.13<br>I(1) 3.52 4.19 5.76 |
| Paraguay | (4, 4, 4, 4, 4, 4) | 7.113 *** | Yes | I(0) 2.41 2.91 4.13<br>I(1) 3.52 4.19 5.76 |
| Peru | (1, 3, 0, 0) | 1.106 | No | I(0) 2.41 2.91 4.13<br>I(1) 3.52 4.19 5.76 |
| Suriname | (4, 4, 4, 4, 4) | 1.828 | No | I(0) 2.41 2.91 4.13<br>I(1) 3.52 4.19 5.76 |
| Uruguay | (4, 1, 4, 4, 4, 4) | 9.722 *** | Yes | I(0) 2.41 2.91 4.13<br>I(1) 3.52 4.19 5.76 |
| Venezuela | (4, 4, 4, 3, 4, 4) | 1.331 | No | I(0) 2.41 2.91 4.13<br>I(1) 3.52 4.19 5.76 |

Notes: This table summarizes the results of the bounds testing approach to co-integration for Equation (8). The critical values for bounds testing are taken from Pesaran, Shin, and Smith (Pesaran et al. 2001, Table CI(iii) Case III, p. 300). **, and *** indicate the statistical significance at the 5 and 1 percent level of significance, respectively.

**Table A5.** Error-Correction Model Estimates.

| Variable | Argentina | Bolivia | Brazil | Chile | Colombia | Costa Rica | Dom. Rep. | Ecuador |
|---|---|---|---|---|---|---|---|---|
| $\vartheta_{t-1}$ | −0.574 *** (0.001) | −0.505 *** (0.000) | −0.855 ** (0.023) | −0.965 ** (0.031) | −0.305 *** (0.001) | −0.373 *** (0.000) | −0.579 *** (0.000) | −0.676 ** (0.016) |
| $\Delta g_{t-1}$ | −0.307 (0.286) | −0.223 (0.458) | 0.304 (0.313) | 1.044 ** (0.044) | 0.408 * (0.076) | 0.608 (0.411) | 1.108 ** (0.041) | −0.234 (0.431) |
| $\Delta g_{t-2}$ | 1.013 *** (0.001) | 1.094 * (0.081) | 0.465 ** (0.013) | 1.208 ** (0.012) | | | 0.007 (0.215) | 0.464 * (0.055) |
| $\Delta g_{t-3}$ | 0.358 * (0.083) | 0.817 (0.118) | 0.360 (0.249) | 0.589 (0.104) | | | | 1.080 (0.138) |
| $\Delta wr_{t-1}$ | 0.106 *** (0.003) | 0.011 (0.445) | 0.012 (0.427) | | −0.011 (0.116) | 0.024 (0.889) | −0.022 (0.400) | 0.019 (0.349) |
| $\Delta wr_{t-2}$ | 0.056 *** (0.004) | 0.035 (0.347) | 0.031 ** (0.017) | 0.067 ** (0.048) | 0.009 (0.812) | 0.015 (0.391) | 0.135 (0.160) | 0.005 (0.928) |
| $\Delta wr_{t-3}$ | 0.062 *** (0.000) | 0.010 (0.442) | 0.020 * (0.073) | 0.103 ** (0.024) | 0.005 (0.522) | 0.027 ** (0.018) | 0.141 * (0.088) | 0.022 (0.585) |
| $\Delta wr_{t-4}$ | −0.065 *** (0.004) | 0.019 ** (0.014) | 0.031 ** (0.020) | −0.056 ** (0.028) | 0.030 (0.261) | −0.014 (0.129) | 0.121 * (0.051) | −0.009 (0.153) |
| $\Delta k_t$ | 0.364 *** (0.002) | 0.507 (0.377) | 0.226*** (0.007) | 0.045 (0.389) | 0.088 * (0.081) | 0.044 (0.129) | 0.192 * (0.056) | 0.335 *** (0.005) |
| $\Delta k_{t-1}$ | 0.307 ** (0.015) | −0.046 (0.494) | 0.117 (0.265) | 0.687 ** (0.037) | 0.052 (0.384) | 0.151 *** (0.004) | 0.360 ** (0.017) | −0.466 * (0.094) |
| $\Delta k_{t-2}$ | −0.275 *** (0.003) | −0.094 (0.241) | 0.045 (0.516) | 0.517 ** (0.021) | 0.099 (0.319) | 0.058 * (0.068) | 0.243 * (0.054) | 0.475 * (0.069) |
| $\Delta k_{t-3}$ | −0.068 *** (0.353) | −0.110 (0.134) | 0.088 (0.122) | 0.105 (0.165) | 0.069 (0.286) | | 0.076 (0.185) | −0.224 (0.135) |
| $\Delta l_t$ | 4.372 *** (0.000) | 2.046 *** (0.000) | 0.047 (0.422) | 5.479 ** (0.023) | 0.104 (0.571) | 0.011 (0.941) | 0.174 (0.517) | 0.533 * (0.084) |
| $\Delta l_{t-1}$ | 5.851 *** (0.002) | 0.163 (0.668) | 0.841 (0.128) | 4.245 ** (0.028) | 0.207 (0.351) | 0.156 (0.262) | 0.083 * (0.096) | 0.245 (0.797) |
| $\Delta l_{t-2}$ | 6.788 *** (0.001) | 0.740 * (0.072) | 0.614 (0.228) | 4.282 ** (0.013) | 0.284 (0.316) | | 0.578 * (0.096) | 0.913 (0.318) |
| $\Delta l_{t-3}$ | 2.551 ** (0.033) | 0.447 (0.174) | 0.632 (0.179) | 1.301 ** (0.017) | 0.216 (0.219) | | 0.774 (0.147) | 1.313 (0.242) |
| $\Delta hc_t$ | −0.342 (0.451) | −0.011 (0.633) | −0.195 (0.273) | −0.158 ** (0.029) | −0.107 (0.202) | 0.011 (0.781) | −0.001 (0.983) | 0.217 (0.182) |
| $\Delta hc_{t-1}$ | −3.235 *** (0.000) | −0.579 (0.333) | −0.107 (0.467) | 0.257 ** (0.043) | 0.026 (0.128) | 0.151 ** (0.010) | 0.013 (0.869) | 0.141 (0.567) |
| $\Delta hc_{t-2}$ | −0.778 ** (0.041) | −0.310 (0.352) | −0.284 * (0.082) | 0.093 (0.182) | 0.157 * (0.064) | | 0.117 (0.419) | 0.252 (0.213) |
| $\Delta hc_{t-3}$ | | −0.030 (0.685) | | −0.242 ** (0.016) | 0.305 (0.179) | | −0.774 (0.147) | −0.302 * (0.087) |
| $\Delta oda_t$ | 0.014 (0.127) | 0.043 (0.271) | 0.018 (0.213) | 0.034 ** (0.014) | 0.016 (0.247) | 0.006 (0.156) | 0.001 (0.983) | 0.016 (0.630) |
| $\Delta oda_{t-1}$ | 0.025 ** (0.028) | 0.083 * (0.081) | 0.044 (0.115) | −0.210 ** (0.028) | −0.002 (0.264) | −0.027 *** (0.000) | −0.014 (0.263) | 0.050 (0.174) |
| $\Delta oda_{t-2}$ | | 0.119 (0.222) | 0.015 (0.430) | −0.050 * (0.077) | 0.014 (0.279) | −0.020 *** (0.000) | −0.009 (0.185) | 0.055 (0.132) |
| $\Delta oda_{t-3}$ | | 0.078 * (0.051) | 0.009 (0.627) | 0.011* (0.058) | 0.027 (0.367) | −0.006 * (0.076) | | |

Note: The figures in parentheses are *p*-values. ***, ** and * indicate the statistical significance at the 1%, 5% and 10% levels, respectively.

**Table A6.** Error-Correction Model Estimates (Continued).

| Variable | El Salvador | Guatemala | Guyana | Mexico | Nicaragua | Panama | Paraguay | Uruguay |
|---|---|---|---|---|---|---|---|---|
| $\vartheta_{t-1}$ | −0.853 ** | −0.817 *** | −0.439 ** | −0.358 ** | −0.553 ** | −0.811 *** | −0.807 ** | −0.580 *** |
| | (0.045) | (0.000) | (0.019) | (0.012) | (0.034) | (0.042) | (0.013) | (0.000) |
| $\Delta g_{t-1}$ | −0.242 | 0.968 *** | 0.030 | 1.745 ** | 1.023 * | 0.578 | 0.552 | 0.206 |
| | (0.604) | (0.002) | (0.862) | (0.028) | (0.059) | (0.515) | (0.523) | (0.133) |
| $\Delta g_{t-2}$ | 0.685 | | 0.379 | 1.159 ** | 0.386 | 1.065 | 0.807 | 0.332 ** |
| | (0.274) | | (0.137) | (0.036) | (0.214) | (0.369) | (0.376) | (0.014) |
| $\Delta g_{t-3}$ | | | | 0.514 | 0.239 | −0.907 | −0.449 | 0.200 ** |
| | | | | (0.126) | (0.214) | (0.450) | (0.369) | (0.049) |
| $\Delta wr_{t-1}$ | 0.134 * | 0.010 ** | 0.018 | 0.440 ** | 0.117 | 0.097 * | 0.008 | −0.025 |
| | (0.075) | (0.032) | (0.174) | (0.023) | (0.197) | (0.081) | (0.819) | (0.182) |
| $\Delta wr_{t-2}$ | −0.057 | −0.010 * | 0.059 ** | 0.232* | 0.101 | −0.174 | 0.088 | |
| | (0.336) | (0.082) | (0.027) | (0.064) | (0.147) | (0.278) | (0.145) | |
| $\Delta wr_{t-3}$ | 0.009 | −0.008 ** | −0.018 | 0.234* | −0.001 | −0.143 | −0.064 | |
| | (0.814) | (0.028) | (0.283) | (0.094) | (0.983) | (0.139) | (0.498) | |
| $\Delta wr_{t-4}$ | −0.049 | −0.004 | | −0.137 | −0.073 | −0.071 | 0.393 | |
| | (0.228) | (0.166) | | (0.121) | (0.571) | (0.271) | (0.271) | |
| $\Delta k_{t}$ | 0.044 | 0.057 | 0.008 | 0.115 | 0.145 | 0.388 | 0.087 | 0.136 *** |
| | (0.452) | (0.289) | (0.715) | (0.456) | (0.111) | (0.192) | (0.630) | (0.002) |
| $\Delta k_{t-1}$ | 0.146 | −0.017 | 0.074 | −0.201 | −0.279 | −0.224 | 0.152 | −0.237 ** |
| | (0.172) | (0.824) | (0.116) | (0.229) | (0.212) | (0.336) | (0.298) | (0.010) |
| $\Delta k_{t-2}$ | 0.071 | −0.068 | 0.015 | −0.460 ** | −0.111 | 0.057 | −0.242 | −0.161 ** |
| | (0.187) | (0.132) | (0.665) | (0.044) | (0.426) | (0.681) | (0.119) | (0.013) |
| $\Delta k_{t-3}$ | | −0.062 | −0.130 ** | −0.451 ** | −0.103 | 0.087 | 0.299 | −0.146 ** |
| | | (0.182) | (0.048) | (0.026) | (0.318) | (0.652) | (0.142) | (0.047) |
| $\Delta l_{t}$ | 0.024 | 0.296 | 0.988 | 3.355 * | 3.636 | 0.654 | 1.096 | 0.798 ** |
| | (0.517) | (0.229) | (0.419) | (0.060) | (0.288) | (0.693) | (0.201) | (0.049) |
| $\Delta l_{t-1}$ | 0.043 | 0.369 | −2.435 | −1.903 | −2.038 | −1.650 | −0.464 | −2.998 *** |
| | (0.930) | (0.565) | (0.122) | (0.151) | (0.121) | (0.202) | (0.356) | (0.001) |
| $\Delta l_{t-2}$ | −0.148 | −1.247 | −0.904 | | −2.913 ** | −3.170 | −2.568 * | −1.587 ** |
| | (0.721) | (0.392) | (0.312) | | (0.015) | (0.361) | (0.098) | (0.019) |
| $\Delta l_{t-3}$ | 0.658 | −1.572 | | | | −2.695 | 0.901 | −1.385 ** |
| | (0.080) | (0.347) | | | | (0.287) | (0.310) | (0.041) |
| $\Delta hc_{t}$ | −0.038 | 0.067 | 0.096 ** | 0.020 | 0.535 | 0.875 | 0.005 | 0.141 *** |
| | (0.387) | (0.243) | (0.025) | (0.786) | (0.457) | (0.417) | (0.920) | (0.000) |
| $\Delta hc_{t-1}$ | 0.088 | 0.074 | −0.010 | 0.661 ** | 1.151 * | 1.630 | 0.087 | 0.222 *** |
| | (0.156) | (0.212) | (0.636) | (0.018) | (0.079) | (0.231) | (0.234) | (0.001) |
| $\Delta hc_{t-2}$ | 0.054 | 0.119 ** | −0.069 ** | 0.411 ** | 0.080 | 1.436 | 0.146 * | 0.111 *** |
| | (0.159) | (0.046) | (0.027) | (0.026) | (0.859) | (0.274) | (0.099) | (0.007) |
| $\Delta hc_{t-3}$ | | 0.062 | | | −0.212 | 1.152 | 0.058 | −0.039 |
| | | (0.328) | | | (0.680) | (0.307) | (0.110) | (0.254) |
| $\Delta oda_{t}$ | −0.006 | 0.017 | −0.010 | 0.026 | −0.099 * | −0.024 | −0.027 | 0.048 *** |
| | (0.782) | (0.259) | (0.750) | (0.161) | (0.050) | (0.603) | (0.386) | (0.001) |
| $\Delta oda_{t-1}$ | 0.018 | 0.021 | 0.053 ** | 0.004 | 0.256 ** | −0.049 | −0.070 | 0.032 ** |
| | (0.527) | (0.188) | (0.026) | (0.726) | (0.026) | (0.551) | (0.323) | (0.017) |
| $\Delta oda_{t-2}$ | 0.011 | 0.029 | 0.028 | −0.059 ** | 0.163 ** | −0.057 | −0.137 | −0.024 * |
| | (0.523) | (0.122) | (0.246) | (0.015) | (0.044) | (0.586) | (0.234) | (0.091) |
| $\Delta oda_{t-3}$ | 0.029 | 0.025 | | −0.080 ** | 0.098 ** | 0.053 | 0.126 | −0.034 *** |
| | (0.129) | (0.243) | | (0.014) | (0.043) | (0.508) | (0.129) | (0.008) |

Note: The figures in parentheses are *p*-values. ***, ** and * indicate the statistical significance at the 1%, 5% and 10% levels, respectively.

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
