# Peer review of "Do Remittances Promote Economic Growth and Reduce Poverty? Evidence from Latin American Countries"

_economies, doi:10.3390/economies8020035_

Round 1

Reviewer 1 Report

This paper makes an attempt to add to the vast literature of remittances and economic growth (and poverty) by putting specific attention to Latin America. The major limitation of this paper is its motivation. The paper didn't provide any discussion related to the existing literature and identify the gaps. Are there any reasons why would we expect the results to be different in this study compared to the existing studies? pointing out those studies would better motivate the analysis. The paper didn't provide any justification to the specifications being used or the advantages of such a specification. Why the model is linear? The overall model is not fully justified. The literature suggest that economic growth not only depends on physical capital, it's likely to be influenced by human capital. The authors could extend the discussion by incorporating human capital into the model. 

The authors have used all governance indicators together. However, the governance indicators are highly correlated. Those indicators should be used interchangeably or the authors could create an index based on all six indicators following the Principal Component Analysis (PCA). The authors need to explain the importance of using FMOLS and ARDL compared to the other methods. While the remittance literature has expanded in various directions, the test of simple relationship between remittances and growth (and poverty) would seem like a graduate exercise. The authors need to provide better justifications and be innovative in terms of examining the impacts of remittances on mitigating various shocks. For example, the authors can estimate a model to capture the non-linearity of remittance flows and how it affects growth overtime. It would be interesting to see how different shocks can influence the impact of remittances on economic growth. This research is a good starting point but it falls short of being innovative/motivated enough. 

Author Response

Reviewer 1 Comments

This paper makes an attempt to add to the vast literature of remittances and economic growth (and poverty) by putting specific attention to Latin America. The major limitation of this paper is its motivation. The paper didn't provide any discussion related to the existing literature and identify the gaps. Are there any reasons why would we expect the results to be different in this study compared to the existing studies? pointing out those studies would better motivate the analysis. The paper didn't provide any justification to the specifications being used or the advantages of such a specification. Why the model is linear? The overall model is not fully justified. The literature suggest that economic growth not only depends on physical capital, it's likely to be influenced by human capital. The authors could extend the discussion by incorporating human capital into the model.

The authors have used all governance indicators together. However, the governance indicators are highly correlated. Those indicators should be used interchangeably or the authors could create an index based on all six indicators following the Principal Component Analysis (PCA). The authors need to explain the importance of using FMOLS and ARDL compared to the other methods. While the remittance literature has expanded in various directions, the test of simple relationship between remittances and growth (and poverty) would seem like a graduate exercise. The authors need to provide better justifications and be innovative in terms of examining the impacts of remittances on mitigating various shocks. For example, the authors can estimate a model to capture the non-linearity of remittance flows and how it affects growth overtime. It would be interesting to see how different shocks can influence the impact of remittances on economic growth. This research is a good starting point but it falls short of being innovative/motivated enough.

Response:

Thank you very much for your valuable comments. We have addressed your concerns and revised the paper accordingly. Here are the specific revisions made to the original manuscript:

We have explained our motivation to conduct this study after introducing a literature review section to the paper. Based on your comments, we experimented with a nonlinear model by introducing square and cubic terms to the model. That resulted in very poor results with incorrect signs and statistically insignificant coefficients. Thus, we found out that a linear model fits the data better. Based on your suggestion, we introduced human capital variable to the model. We re-estimated the models using one governance indicator at a time instead of using all of them together. We have explained the reason for the use of FMOLS and ARDL methods.

Based on comments by another reviewer, we have also made the following change to the original manuscript:

We have introduced relevant information from Latin American countries about GDP and poverty.

Reviewer 2 Report

The paper is interesting, especially recently there is revived interest in the problem of remittances and their linkages with macroeconomic fundamentals.
The methods used are adequate to the objective and allow to answer the main objectiv under study.
My general opinion about this paper is positive. Neverthless, I believe that some key aspects could be improved:
- it would be advisable to introduce relevant information from Latin American countries and about GDP and poverty (only remittances are addressed)
- in line 75 it is spoken of the literature review ”The next section presents a survey of the literature”, but I did not find such a section in the paper, although this is necessary especially given the numerous investigations tackling exactly this topic;
- the authors could provide a stronger link between the evidence obtained and previous results on the same topic;
- there is a list of references, but they do not have correspondence in the text (lines: 382, 387, 389, 391, 398, 404, 406, 410, 415, 418)

Author Response

Reviewer 2 Comments

The paper is interesting, especially recently there is revived interest in the problem of remittances and their linkages with macroeconomic fundamentals.

The methods used are adequate to the objective and allow to answer the main objectiv under study.

My general opinion about this paper is positive. Neverthless, I believe that some key aspects could be improved:

- it would be advisable to introduce relevant information from Latin American countries and about GDP and poverty (only remittances are addressed)

- in line 75 it is spoken of the literature review ”The next section presents a survey of the literature”, but I did not find such a section in the paper, although this is necessary especially given the numerous investigations tackling exactly this topic;

- the authors could provide a stronger link between the evidence obtained and previous results on the same topic;

- there is a list of references, but they do not have correspondence in the text (lines: 382, 387, 389, 391, 398, 404, 406, 410, 415, 418)

Response:

Thank you very much for your valuable comments. We have addressed your concerns and revised the paper accordingly. Here are the specific revisions made to the original manuscript:

We have introduced relevant information from Latin American countries about GDP and poverty. We have included a Literature Review section to the paper. Based on the literature review, we have compared our findings with that of previous studies. We have included the missing references.

Based on comments by another reviewer, we have also made following changes to the original manuscript:

We have explained our motivation to conduct this study after introducing a literature review section to the paper. We have introduced human capital variable to the model. We re-estimated the models using one governance indicator at a time instead of using all of them together.

Reviewer 3 Report

This paper covers a very important topic, with strong policy implications.

The methodology of the paper is adequate but it suffers from some important drawbacks that must be addresed.

I emphasize the following aspects:

please separate the introduction from a background section. In the context of this process, please include Tables 1 and 2 in the background. in fact, some additional discussion could be useful in order to provide a stronger support for the empirical exercise conducted in the paper; the intepretation of the evidence must be strongly revised, namely through the inclusion of references with previous results in order to compare them with the results obtained here; In general terms, I feel that the paper is poor in terms of link to previous literature. Additional discussion on the background section and the empirical part could help to solve this problem.

Author Response

Reviewer 3 Comments

This paper covers a very important topic, with strong policy implications.

The methodology of the paper is adequate but it suffers from some important drawbacks that must be addresed. I emphasize the following aspects:

please separate the introduction from a background section. In the context of this process, please include Tables 1 and 2 in the background. in fact, some additional discussion could be useful in order to provide a stronger support for the empirical exercise conducted in the paper; the intepretation of the evidence must be strongly revised, namely through the inclusion of references with previous results in order to compare them with the results obtained here; In general terms, I feel that the paper is poor in terms of link to previous literature. Additional discussion on the background section and the empirical part could help to solve this problem.

Response:

Thank you very much for your valuable comments. We have addressed your concerns and revised the paper accordingly. Here are the specific revisions made to the original manuscript:

We have separated the introduction from background section. We have included a Literature Review section to the paper. Based on the literature review, we have compared our findings with that of previous studies.

Based on comments by another reviewer, we have also made following changes to the original manuscript:

We have explained our motivation to conduct this study after introducing a literature review section to the paper. We have introduced human capital variable to the model. We re-estimated the models using one governance indicator at a time instead of using all of them together.

Round 2

Reviewer 3 Report

The new version of the paper answers my previous concerns. Therefore, in my opinion, the paper can be accepted for publication.

Author Response

Thank you for your valuable comments. Please find attached the revised paper.
